# Stay-Positive: A Case for Ignoring Real Image Features in Fake Image Detection

Anirudh Sundara Rajan [1]    Yong Jae Lee [1]

## Abstract

Detecting AI-generated images is a challenging yet essential task. A primary difficulty arises from the detector's tendency to rely on spurious patterns, such as compression artifacts, which can influence its decisions. These issues often stem from specific patterns that the detector associates with the real data distribution, making it difficult to isolate the actual generative traces. We argue that an image should be classified as fake if and only if it contains artifacts introduced by the generative model. Based on this premise, we propose Stay-Positive, an algorithm designed to constrain the detector's focus to generative artifacts while disregarding those associated with real data. Experimental results demonstrate that detectors trained with Stay-Positive exhibit reduced susceptibility to spurious correlations, leading to improved generalization and robustness to post-processing. Additionally, unlike detectors that associate artifacts with real images, those that focus purely on fake artifacts are better at detecting inpainted real images. For implementation details, please visit: https://anisundar18.github.io/Stay-Positive.

## 1. Introduction

Recent advancements in deep generative modeling, particularly diffusion models (Sohl-Dickstein et al., 2015; Song & Ermon, 2019) and flow-based models (Liu et al., 2023; Lipman et al., 2022), have significantly improved image generation capabilities. The combination of large-scale datasets (Schuhmann et al., 2022) and architectural innovations (Dhariwal & Nichol, 2021; Esser et al., 2024) has enabled modern systems to generate high-quality images (Pernias et al., 2023; Labs; Podell et al., 2024), often conditioned on auxiliary inputs such as text. While these ad-

vancements have expanded the potential applications of generative models, they also raise concerns about misuse, including misinformation and fraud. Image forensics aims to address these risks by developing methods to reliably detect AI-generated images.

Significant progress has been made in fake image detection, with detectors trained to generalize across CNN-based generators (Wang et al., 2020) and extended to unseen architectures (Ojha et al., 2023). However, reliably detecting fake images from a known generator family remains a challenge. A key difficulty arises when images undergo post-processing (e.g., compression), as demonstrated by Rajan et al. (2025), where detectors trained on images generated by latent diffusion models (Vahdat et al., 2021; Rombach et al., 2022) exhibited reduced effectiveness in detecting post-processed images. Specifically, detectors trained by Corvi et al. (2023) and Rajan et al. (2025) were found to spuriously associate WEBP compression artifacts with real images, which negatively impacted their performance. In this work, we focus on improving the detection of fake images generated by a known model family by mitigating the impact of spurious correlations. The most effective approach for developing a fake image detector is to train a neural network-based binary classifier. However, its performance is highly dependent on the training data, and any discriminative feature associated with the data, including subtle post-processing artifacts, can influence the detector's decisions. A common source of such issues is the use of real images for training, which are often collected from online platforms and may have undergone unknown operations such as compression and resizing prior to upload. As a result, detectors can learn spurious correlations. Such issues with compression artifacts were also observed in the widely adopted GenImage benchmark (Zhu et al., 2024), as reported by Grommelt et al. (2024).

In addition to robustness against post-processing, it is essential for detectors to accurately identify images generated by newer models within the same generator family. We observe that reliance on spurious correlations hinders generalization to such models. During training, detectors often focus on differences in image quality between real and fake images, leading them to associate certain artifacts with real images. However, such hypotheses are often incorrect, as the same artifacts can appear in fake images generated by improved models within the same family. This reliance on

[1]University of Wisconsin-Madison. Correspondence to: Anirudh Sundara Rajan <asundararaj2@wisc.edu>.

*Proceedings of the 42$^{nd}$ International Conference on Machine Learning*, Vancouver, Canada. PMLR 267, 2025. Copyright 2025 by the author(s).

such artifacts limits the detector's ability to generalize to newer generators. For instance, a detector trained on images generated by Latent Diffusion Models (LDM) (Rombach et al., 2022) (Corvi et al., 2023) struggles to generalize to images generated by FLUX (Labs) for this very reason.

Our main idea is that images generated by a specific generator family should contain distinct artifacts. An ideal detector should focus exclusively on these fake artifacts, with their absence indicating that the image does not originate from a generator of that family. Relying on artifacts associated with real images is, at best, unnecessary and, at worst, misleading. We show detectors perform better when real image features do not affect decisions.

To identify features linked to real images, we make the following assumptions: the output of the entire network passes through a ReLU activation before the final layer, and the final output is passed through a sigmoid activation for binary classification, where class 1 represents fake images and class 0 represents real images. Under these conditions, we show that features connected to negative weights in the final layer correspond to patterns found in real images. To mitigate the influence of these features, we propose a simple yet highly effective algorithm that retrains the last layer to minimize the loss while ensuring that the weights remain strictly non-negative. This adjustment forces the detector to make its decision using only fake image patterns, ignoring spurious real image artifacts. As a result, the detector shows improved generalization. This approach also enables our detectors to effectively detect partially inpainted real images, unlike conventional detectors, which may be influenced by real-image features. We validate our method by improving the performance of detectors from Corvi et al. (2023) and Rajan et al. (2025). Ultimately, we hope our findings contribute to the community's broader efforts to combat misinformation.

## 2. Background

In this section, we first define the problem, describe the general details of training a fake image detector, and introduce the notation used throughout the paper.

### 2.1. Problem Definition

The task of *fake image detection* is a binary classification problem. Given a dataset $\mathcal{D} = \{(\mathbf{x}_i, y_i)\}_{i=1}^{N}$ of $N$ labeled samples, where each $\mathbf{x}_i \in \mathbb{R}^{H \times W \times C}$ represents an image of height $H$, width $W$, and $C$ channels, and $y_i \in \{0, 1\}$ is the corresponding label, the goal is to learn a mapping $f : \mathbb{R}^{H \times W \times C} \to \{0, 1\}$. The label $y_i = 0$ denotes a *real* image, while $y_i = 1$ indicates a *fake* image generated by a neural network.

### 2.2. Learning-Based Fake Image Detection

Given a known generative model family of interest, the well-established method of training a fake image detector involves the following steps. First, a set of real images, usually sourced online, is selected, while fake images are typically generated using the targeted generative model. A neural network detector $f_\theta$ is then trained to solve the binary classification task.

In our work, we assume the following neural network structure for $f_\theta$. The network is composed of three components: (i) a feature extraction network, $g_\phi : \mathbb{R}^{H \times W \times C} \to \mathbb{R}^d$, which encodes the input image $\mathbf{x}$ into a $d$-dimensional feature vector, where $\phi$ represents the parameters of the feature extraction network; (ii) a ReLU activation function, applied element-wise to the extracted features, denoted by $\gamma$; and (iii) a linear classifier, parameterized by $\mathbf{w} \in \mathbb{R}^d$ and $b \in \mathbb{R}$, followed by a sigmoid activation function. The parameters of the network are $\theta = \{\phi, \mathbf{w}, b\}$.

Formally, the output of $f_\theta$ is given by

$$f_\theta(\mathbf{x}) = \sigma(\mathbf{w}^\top \gamma(g_\phi(\mathbf{x})) + b),$$

where $\sigma(\cdot)$ denotes the sigmoid function. This network is trained using the binary cross entropy loss.

## 3. Issues with Associating the Presence of Specific Features with Real Images

In this section, we first examine how unknown post-processing artifacts in real training images negatively impact detector performance. We then address the broader issues associated with linking specific artifacts to real images.

### 3.1. Case Study 1: Post-Processing Artifacts

Detectors can inadvertently let subtle differences between real and fake training distributions, such as spurious features like compression or resizing artifacts, influence their decision. Real images, often sourced from online platforms, may have undergone unknown post-processing, making it difficult to determine which features the detector may associate with real images. We explore this issue below.

In their study on fake image detection for latent diffusion models, Corvi et al. (2023) and Rajan et al. (2025) trained a ResNet-50 (He et al., 2016) using real images from the LSUN (Yu et al., 2016) and COCO (Lin et al., 2015) datasets. The LSUN images were compressed using the WEBP algorithm but saved in the lossless PNG format by the dataset creators. The fake images used to train the detectors did not contain these WEBP artifacts, leading the detectors to associate WEBP compression with the real distribution. We demonstrate this issue in the following experiment.

**Experiment Details**: We aim to test whether the detector

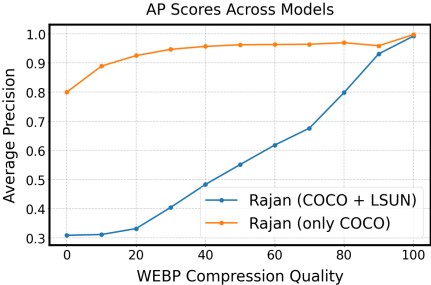

*Figure 1.* **Sensitivity to WEBP Compression.** Using the LSUN dataset, which contains WEBP compressed images, as part of the real distribution makes the network highly vulnerable to WEBP compression.

confuses WEBP-compressed fake images with real images, and if this confusion is driven by the inclusion of LSUN images in the real distribution. To do so, we adopt the experimental setup proposed by Rajan et al. (2025), where fake images are generated by reconstructing real images through the LDM (Rombach et al., 2022) autoencoder. We train two detectors: one that includes LSUN images in the real distribution and another that excludes them, training only on COCO images. For testing, we use 500 real images from the Redcaps dataset (Desai et al., 2021) and 500 fake images generated by Stable Diffusion 1.5. Various levels of WEBP compression are applied to the fake images, while the real images remain uncompressed. We evaluate the average precision (AP), where a drop in AP with increasing compression levels would suggest that the detector is confusing WEBP-compressed fake images with real ones.

**Analysis**: The results of the experiment are shown in Fig. 1. In the training data, LSUN images are WEBP compressed, while the fake images lack WEBP artifacts. This dataset imbalance leads the detector to associate WEBP compression with real images, as evidenced by the drop in AP as WEBP Compression Quality decreases (i.e., there is more WEBP compression). Excluding the LSUN images mitigates this issue. However, such details will not always be known, making it challenging to filter the data effectively. Thus, an algorithm is needed to eliminate spurious correlations associated with the real distribution.

### 3.2. Case Study 2: Beyond Post-Processing Artifacts

We next argue that any pattern the detector associates with the real distribution could be spurious. We begin by explaining our intuition.

**Hypothesis:** Consider a detector trained on real images versus LDM-generated fake images. Models based on 4-channel autoencoders, like LDM, struggle to reconstruct fine details in real images, such as text, as noted by prior work (Dai et al., 2023). As a result, the detector may associate the

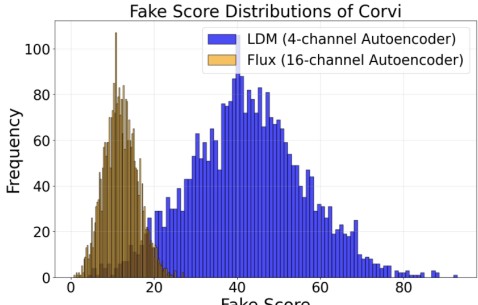

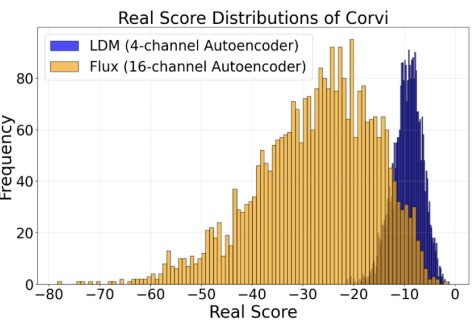

*Figure 2.* **Image Quality-Based Spurious Features.** *Corvi* outputs a higher real score for Flux reconstructions compared to LDM reconstructions demonstrating the spurious nature of these real features. Fake Score reduces due to the use of a different generator.

presence of certain fine details with real images. However, this hypothesis does not hold, as Dai et al. (2023) also show that 16-channel autoencoder-based models, such as FLUX, can reconstruct such details, indicating these features do not determine if an image is real. We highlight a similar issue with *Corvi* (Corvi et al., 2023).

**Real and Fake Score:** We aim to demonstrate that artifacts the detector associates with real images can also appear in fake images from the same generator family, influencing the detector's decision. To achieve this, we first develop a method to measure the impact of these artifacts on the detector's decision.

Consider a trained fake image detector as described in Section 2.2. Given an arbitrary fake image $\mathbf{x}$, we define the extracted feature as $\mathbf{h} = \gamma(g_\phi(\mathbf{x})), \quad \mathbf{h} \in \mathbb{R}^d_{\geq 0}$. Where $\gamma$ is the ReLU activation. Our final network output can be written as follows,

$$f_\theta(\mathbf{x}) = \sigma\Big( \underbrace{\sum_{\mathbf{w}_i > 0} \mathbf{w}_i \mathbf{h}_i}_{\text{Increases sum, fakeness}} + \underbrace{\sum_{\mathbf{w}_i < 0} \mathbf{w}_i \mathbf{h}_i}_{\text{Decreases sum, realness}} + b \Big).$$

(1)

We assume that a fake image $\mathbf{x}$ is associated with the label 1 whereas a real image has the label 0. We also assume that the final score (added with the bias) is passed through

**1** **Train a Fake Image Detector**

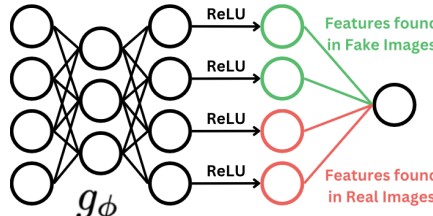

**2** **Re-train the last layer**

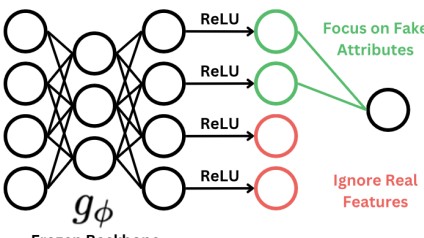

*Figure 3.* Our key idea involves 2 steps. (1) We first train a fake image detector in the standard way without any modifications. This detector focuses on both real and fake features. (2) We re-train the last layer of the network such that it only focuses on the fake features to make a decision.

a monotonically increasing sigmoid function, therefore we can infer that the final score, represented as $\mathbf{w}^T\mathbf{h}$, for an ideal detector must be higher for a fake image than for a real image. Our use of the ReLU activation ensures that the extracted feature vector $\mathbf{h}$ is a vector with only non-negative values. Based on this, for a dimension $i$, if $\mathbf{w}_i < 0$, we can conclude that the weighted contribution $\mathbf{w}_i\mathbf{h}_i \leq 0$. This operation reduces the final score, thus making the image more likely to be classified as real. Similarly, if $\mathbf{w}_i > 0$, the presence[1] of this feature $\mathbf{h}_i$ increases the likelihood of the image being classified as fake. Based on this, we can define a score which quantifies the presence of real and fake features in an image.

$$\text{Real Score} = \sum_{\mathbf{w}_i < 0} \mathbf{w}_i\mathbf{h}_i, \quad \text{Fake Score} = \sum_{\mathbf{w}_i > 0} \mathbf{w}_i\mathbf{h}_i.$$

**Experiment Details:** *Corvi* (trained on LDM-generated images) struggles with detecting FLUX.1-dev (Labs) generated images. To investigate the cause, we use 3,000 real images from the COCO dataset, which were also part of *Corvi*'s training. Since *Corvi* was trained on LDM-generated images, we reconstruct these images using the autoencoders of both LDM and FLUX.1-dev. Importantly, both LDM and FLUX reconstructions have the same semantic content, differing only in autoencoder capability. Our aim is to show that *Corvi* assigns a higher "real" score to FLUX-generated images compared to LDM-generated images, indicating that features associated with real images are also present in fake images generated by FLUX, which belongs to the same generator family as LDM.

**Analysis**: The results are presented in Fig. 2. The real scores of FLUX-generated images (bottom plot) are often higher in magnitude than those of LDM-generated images. This is problematic, as it suggests that features associated with real images by *Corvi* are also present in fake im-

[1]**presence** of a feature refers to $h_i$ having a non-zero value, as opposed to absence where it would have a 0 value.

ages generated by FLUX. Furthermore, compared to LDM-generated fake images, FLUX-generated images have significantly lower fake scores (top plot). As a result, the majority of decisions for FLUX-generated images are influenced by the presence of real features, supporting our argument.

## 4. Stay-Positive: Learning to Ignore Real Features

In Section 3, we showed that associating specific patterns with the real distribution can undermine the detector's effectiveness. Building on this, we argue that an image should be classified as fake if it contains artifacts linked to the generative model of interest, while the absence of these artifacts indicates that the image is real. Consequently, we require an algorithm that forces the detector to focus solely on the patterns associated with the fake distribution. This intuition is illustrated in Figure 3.

**Real and Fake Features:** Fundamentally, the presence of a real feature in an image should increase the likelihood of the detector classifying the image as real. In Section 2.2, under certain assumptions, we demonstrated that the sign of the weights corresponding to each feature can be used to determine whether the presence of that feature enhances the probability of an image being classified as real. Specifically, the indices of $\mathbf{h}$ that are multiplied by positive values of $\mathbf{w}$ correspond to fake features $\mathcal{I}_{\text{fake}}$, while the indices multiplied by negative values of $\mathbf{w}$ correspond to real features $\mathcal{I}_{\text{real}}$. We formalize this below,

$$\mathcal{I}_{\text{real}} = \{i \mid \mathbf{w}_i < 0\}, \quad \mathcal{I}_{\text{fake}} = \{i \mid \mathbf{w}_i \geq 0\}.$$

This perspective allows us to train the detector to focus only on the features that distinguish fake images.

**Stay-Positive to Ignore Real Features:** Based on our analysis in Section 3.2, we aim for the real score to be 0 for all images. By definition, avoiding negative connections in the last layer guarantees this outcome. We achieve this

**Algorithm 1** Re-training the last-layer while staying positive

> **Input:** Pretrained $g_\phi$, learning rate $\eta$, iterations $T$
> Initialize $\mathbf{w} = \mathbf{0}$, $b = 0$
> Freeze $g_\phi$
> **for** $t = 1$ **to** $T$ **do**
> $\quad$ Sample $\{\mathbf{x}_i, y_i\}_{i=1}^{B}$
> $\quad$ $\mathbf{h}_i = \gamma(g_\phi(\mathbf{x}_i))$ $\qquad\qquad$ # where $\gamma$ denotes ReLU
> $\quad$ $f_\theta(\mathbf{x}_i) = \sigma(\mathbf{w}^\top \mathbf{h}_i + b)$
> $\quad$ $\mathcal{L} = -y_i \log(f_\theta(\mathbf{x}_i)) - (1 - y_i) \log(1 - f_\theta(\mathbf{x}_i))$
> $\quad$ $\mathbf{w} \leftarrow \mathbf{w} - \eta \nabla_{\mathbf{w}} \mathcal{L}$
> $\quad$ $b \leftarrow b - \eta \nabla_b \mathcal{L}$
> $\quad$ $\mathbf{w} \leftarrow \max(\mathbf{w}, \mathbf{0})$
> **end for**
> **Output:** $\mathbf{w}, b$

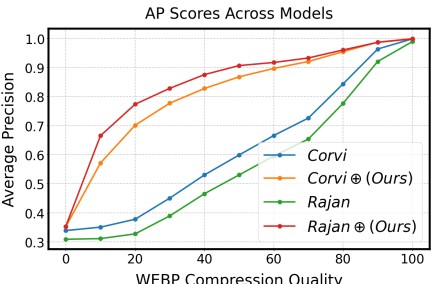

Figure 4. **Improved Robustness to WEBP Compression.** Compared to the original *Corvi* and *Rajan*, our detectors *Corvi⊕* and *Rajan⊕* show increased robustness towards WEBP Compression.

by constraining the last-layer weights during optimization, overwriting any negative values with zero. This ensures that the network relies solely on $\mathcal{I}_{\text{fake}}$ to fit the training data. In our experiments, we found that training the entire network with this constraint negatively impacted the classification accuracy of real images. We report these results in Section 5.5. We hypothesize that when the entire network is trained, it may learn to link the absence of spurious real features to fake images, using negations. We discuss this further in Appendix A.6. To avoid this, we perform last-layer re-training instead. The details are outlined in Algorithm 1. The algorithm illustrates a simple SGD update, though it is independent of the optimizer used.

We use validation accuracy to select the model and determine the stopping condition, as outlined in Appendix A.1.

## 5. Experiments

In this section, we evaluate whether ignoring real features can improve the performance of existing detectors. We consider the following state-of-the-art baselines: The *Corvi* detector (Corvi et al., 2023), which is based on a ResNet-50 network trained on real images from MSCOCO and LSUN, and fake images generated by LDM with prompts corresponding to the real data. Similarly, the *Rajan* detector (Rajan et al., 2025) uses the same real images as *Corvi*, but instead trains on fake images generated by LDM reconstructions of the real images.

Both detectors are trained using the best practices suggested by Gragnaniello et al. (2021), with details provided in Appendix A.1. We apply Algorithm 1 to both detectors while using the same datasets as *Corvi* and *Rajan*. The resulting detectors are referred to as *Corvi⊕ (Ours)* and *Rajan⊕ (Ours)*. We also conduct some of these experiments with detectors trained on GAN-generated images, and these results are included in Appendix A.7.

### 5.1. Mitigating Post-Processing based Spurious Correlations

#### 5.1.1. COMPRESSION-BASED ARTIFACTS

Both *Corvi* and *Rajan* suffer from the spurious correlations studied in Section 3.1, where WEBP compressed images are detected as real images.

**Experiment Details**: To study the sensitivity of *Corvi⊕ (Ours)* and *Rajan⊕ (Ours)* to WEBP compression we utilize the same setting used in Section 3.1.

**Analysis**: We report the results in Figure 4. Compared to *Corvi* and *Rajan*, our detectors are much more robust to WEBP compression. Additionally, our detectors, while ignoring real features, are still able to separate generated images from real images, as indicated by the AP scores. It is important to note that our solution does not require any prior knowledge about specific spurious features, unlike data augmentation-based solutions.

#### 5.1.2. RESIZING-BASED ARTIFACTS

*Corvi* struggles with downsized fake images, as noted by Rajan et al. (2025). This issue arises from the data augmentation strategy used during training, where real images, which typically have higher resolutions, are randomly cropped and downsized to 256×256. This augmentation causes the detector to associate downsizing with real images. We also repeat this experiment using images from the Synthbuster dataset (Cozzolino et al., 2024), details of which can be found in Appendix A.3.

**Experiment Details**: With this experiment, we want to compare the robustness of *Corvi⊕ (Ours)* with the original *Corvi* with respect to downsizing. To do so, we randomly select 500 real images from *whichfaceisreal* (whi) and generate fake images using SDv2.1. All images have a resolution of 1024x1024. We downsize the fake images to different scales and plot AP vs. the scaling factor.

**Analysis**: We plot AP with respect to the scaling factor in

*Table 1.* **AP of existing detectors before and after last-layer retraining.** The large improvements on FLUX and aMUSEd suggest that existing detectors are harmed by spurious correlations unrelated to post-processing, which our algorithm circumvents. Additionally, on settings where *Corvi*, *Rajan* already perform well, our method performs the same if not slightly better. AVG is the equally weighted AP average across all generators.

| METHOD | SD | MJ | KD | PG | PIXART | LCM | FLUX | WUERSTCHEN | AMUSED | AVG |
|---|---|---|---|---|---|---|---|---|---|---|
| CORVI | 99.97 | 99.46 | 99.98 | 97.42 | **99.99** | 99.94 | 57.25 | **100** | 89.18 | 94.23 |
| CORVI⊕ (OURS) | **99.98** | **99.79** | **99.99** | **99.99** | **99.99** | **99.99** | **99.33** | 99.99 | **99.80** | **99.88** |
| RAJAN | 99.89 | 99.90 | 99.98 | 99.94 | **100** | 99.99 | 80.64 | 95.13 | 87.20 | 96.22 |
| RAJAN⊕ (OURS) | **99.99** | **99.93** | **99.99** | **99.99** | 99.99 | **99.99** | **90.50** | **97.99** | **98.11** | **98.65** |

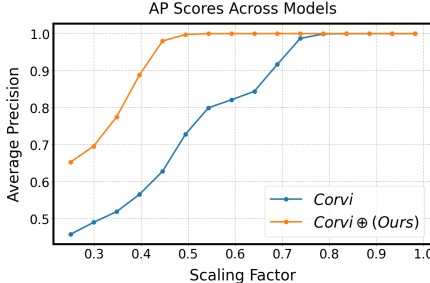

AP Scores Across Models

*Figure 5.* **Improved Robustness to Downsizing.** Compared to the original *Corvi*, our *Corvi⊕* shows increased robustness towards downsampling.

Figure 5. By training on the same dataset, *Corvi⊕ (Ours)* is much more robust to downsizing operations in comparison to *Corvi*. Similar to the case of WEBP compression, we are able to mitigate a spurious correlation without using information about the cause.

### 5.2. Mitigating other Spurious Correlations

Section 3.2 discussed how real features can affect generalization. In this section, we test if *Corvi⊕ (Ours)* and *Rajan⊕ (Ours)* improve over the original versions.

#### 5.2.1. DATASET

To test our hypothesis, we use the following dataset: **i. Real**: 6,000 images from Redcaps (Desai et al., 2021), WikiArt (wik), LAION-Aesthetics (**?**), and whichfaceis-real (whi), including 3,000 post-processed images. **ii. SD**: 3,000 images from SDv1.5, InstructPix2Pix (Brooks et al., 2023), and Nights (Fu et al., 2023); **iii. MJ**: 3,000 images from MidJourney v4 and v5 (mid); **iv. Kandinsky**: 3,100 images from Kandinsky 2.1 (Razzhigaev et al., 2023); **v. PG**: 3,150 images from Playground v2.5 (Li et al., 2024); **vi. PixArt**: 3,150 images from PixelArt-$\alpha$ (Chen et al., 2024b); **vii. LCM**: 3,146 images from Simian-Luo/LCM_Dreamshaper_v7 (Luo et al., 2023); **viii. FLUX**: 3,000 images from Black-Forest-Labs/FLUX.1-dev (Labs); **ix. Wuerstchen**: 3,150 images from warp-ai/wuerstchen

(Pernias et al., 2023); **x. aMUSEd**: 3,150 images from Amused/amused-512 (Patil et al., 2024).

The dataset includes the latest latent space models: aMUSEd (autoregressive) and others (diffusion/flow-based). We generate images for FLUX, Wuerstchen, and aMUSEd, while the rest come from Rajan et al. (2025)'s test set. The real image set covers scenery, art, and faces, with post-processing (compression, resizing, color jitter) to ensure diversity. We demonstrate that our selected real image dataset, represents a wide range of real image types in Appendix A.2. We aim to show that base detectors, *Corvi* and *Rajan*, also learn spurious real features unrelated to post-processing, harming performance. To ensure our experimental results are free from these effects, we use fake images that haven't undergone any post-processing.

#### 5.2.2. DISCUSSION

We present the AP values for the original *Corvi* and *Rajan*, alongside our versions incorporating last-layer retraining, in Table 1. Our methods show substantial improvements in AP on FLUX-generated images. *Corvi⊕ (Ours)* and *Rajan⊕ (Ours)* outperform the original detectors by 42.08 and 9.86, respectively, on FLUX, and by 10.62 and 10.91, respectively, on aMUSEd. As our fake images lack post-processing artifacts, the performance gap is likely due to the spurious correlations discussed in Section 3.2. These improvements suggest that while the original detectors can identify features distinguishing fake images from FLUX and aMUSEd, learning real features hinders the detection of these fake images. Importantly, in settings where *Corvi* and *Rajan* perform well (e.g., SD, KD, LCM), our method matches or outperforms them, showing that the detector does not lose performance by ignoring real features. The rightmost column reports the mean average precision aggregated over all settings.

### 5.3. Comparison with State-of-the-Art Fake Detectors

Next, we compare the performance of our detector with state-of-the-art fake detection methods. We use the same dataset from Section 5.2, but to simulate a real-world set-

ting, we create post-processed versions of the fake images. For FLUX, Wuerstchen, and aMUSEd, we randomly apply compression, resizing, and color jittering, and add these modified images to the dataset. Images for the other models are taken from the dataset provided by Rajan et al. (2025). Additionally, we evaluate our detector on the widely used GenImage benchmark (Zhu et al., 2024), with results presented in Appendix A.4, as well as on latent diffusion and autoregressive models from the UFD benchmark (Ojha et al., 2023), with results shown in Appendix A.5.

### 5.3.1. BASELINES

From the family of zero-shot fake image detectors we select **i. AEROBLADE** (Ricker et al., 2024). From the CLIP linear-probing paradigm, we select: **ii. UFD-ProGAN** (Ojha et al., 2023): trained on ProGAN images, representing the baseline method in this paradigm; **iii. UFD-LDM** (Ojha et al., 2023): trained on LDM images, extending the original setup to a different generative domain; **iv. ClipDet**: a follow-up work by Cozzolino et al. (2024) that refines the linear-probing approach for LDM images. Additionally, we include **v. DRCT**: a recent method by Chen et al. (2024a) that employs DDIM inversion to reconstruct both real and fake images, combined with a contrastive training objective for better generalization, using a ConvNext (Liu et al., 2022) backbone trained on images from SDv1.4. We also compare against the original **Corvi** and **Rajan**, the detectors on which our method is based, to benchmark the improvements our approach introduces.

### 5.3.2. DISCUSSION

We present our results in Table 2 and observe that detectors relying on full-network fine-tuning, such as DRCT, *Corvi*, and *Rajan*, along with our improved baselines, outperform CLIP-based and zero-shot approaches. In comparison to *Corvi*, our *Corvi⊕ (Ours)* shows large improvements. On PG and FLUX generated images, our version outperforms the original *Corvi* by 6.47 and 19.66 respectively. On FLUX generated images, our version outperforms the original *Rajan* by 4.05. Notably, the AP scores of the original *Corvi* and *Rajan* on FLUX and aMUSEd improve compared to the values reported in Table 1, indicating that post-processed fake images from these generators were actually easier to detect compared to fake images without post-processing. This suggests that the fake images used in Table 1 contain artifacts that harm the performance of *Corvi* and *Rajan*. Post-processing these images can remove these artifacts, improving both detectors' performance. Given their sensitivity to post-processing, these are likely low-level artifacts affecting detector performance, similar to those described in Section.3.2. Unlike the original *Corvi* and *Rajan*, our versions are unaffected by such discrepancies.

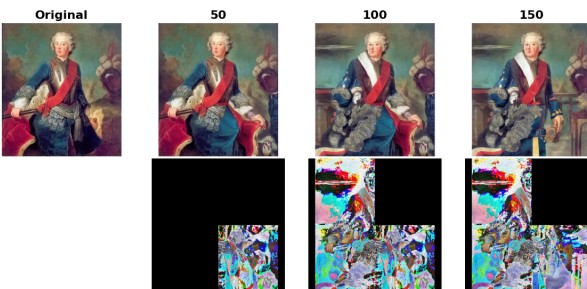

*Figure 6.* **Example of an image which has been recursively inpainted.** The first/second row shows the inpainted image and the inpainted region, respectively.

## 5.4. Improved Detection of Partially Inpainted Images

Prior experiments consider fake images that are completely generated. However, a user can take real images and partially modify them. While most regions of such images are "real", they could be created with malicious intent, making their detection important. Intuitively, detectors that depend on real features would struggle to detect these modified images, since a portion of these images is real. Here, we study the sensitivity of our approach to such images.

### 5.4.1. EXPERIMENT DETAILS

We use the Stable Diffusion inpainted dataset from Conde et al. (2024), where a real image is modified by masking and inpainting a region. Following Conde et al. (2024), we group images by the percentage of inpainted pixels. In recursive inpainting (refer Fig 6), the same region can be inpainted multiple times, so the number of inpainted pixels may exceed the total (e.g., 150%). Each group has 300 inpainted images. We calculate AP with respect to the 6000 real images from Section 5.2.

### 5.4.2. ANALYSIS

We report our results in Table 3. At the 50-level, all approaches barring *Corvi⊕* and *Rajan⊕* fail to detect these inpainted images. This is perhaps unsurprising since these images are "real" for the most part. Our detectors on the other hand, do not suffer from these issues as indicated by the AP score. The performance of *Corvi* and *Rajan* improves with more inpainting, highlighting the harmful impact of real features on their performance.

## 5.5. Ablations

In this section, we ablate other possible alternatives to final-layer re-training. We experimented with two other variants, (i) **+ clamped (no retrain)** where we clamp the detector's weights to stay-positive without re-training the whole network and (ii) **+ clamped (retrain)**: where we re-train the whole network with the stay-positive algorithm applied on

*Table 2.* **AP of state-of-the-art detectors on post-processed fake images.** Methods trained by our algorithm, *Corvi⊕* and *Rajan⊕* outperform their base detectors as well as other state-of-the-art detection approaches on recent generators such as FLUX and Wuerstchen. Additionally, on settings where *Corvi*, *Rajan* already perform well, our method performs the same if not slightly better. AVG is the equally weighted AP average across all generators.

| Method | SD | MJ | KD | PG | PixArt | LCM | FLUX | Wuerstchen | aMUSEd | AVG |
|---|---|---|---|---|---|---|---|---|---|---|
| AEROBLADE (Ricker et al., 2024) | 90.81 | 96.48 | 94.03 | 71.53 | 87.84 | 89.99 | 60.34 | 85.93 | 88.39 | 85.03 |
| UFD-ProGAN (Ojha et al., 2023) | 61.93 | 61.72 | 74.88 | 70.23 | 69.81 | 70.86 | 37.54 | 86.30 | 88.84 | 64.73 |
| UFD-LDM (Ojha et al., 2023) | 62.02 | 52.33 | 65.33 | 62.36 | 62.39 | 65.52 | 35.18 | 86.46 | 90.98 | 69.12 |
| ClipDet (Cozzolino et al., 2024) | 71.63 | 73.72 | 74.71 | 75.53 | 76.61 | 72.06 | 81.48 | 90.11 | 87.61 | 78.16 |
| DRCT (Chen et al., 2024a) | 96.09 | 90.74 | 95.86 | 83.83 | 78.39 | 88.43 | 76.60 | 85.40 | 87.87 | 87.02 |
| Corvi (Corvi et al., 2023) | 97.87 | 94.81 | 95.38 | 91.40 | 94.45 | 96.33 | 74.57 | 95.83 | 94.57 | 92.80 |
| Corvi⊕ (Ours) | 98.94 | 94.92 | 97.71 | 97.87 | 98.59 | 98.73 | **94.23** | **98.16** | 95.47 | 97.17 |
| Rajan (Rajan et al., 2025) | **99.40** | **98.30** | 98.18 | **98.62** | 98.64 | **99.79** | 87.80 | 94.51 | 95.38 | 96.73 |
| Rajan⊕ (Ours) | 99.22 | 96.98 | **98.22** | 98.53 | **99.11** | 99.57 | 91.85 | 94.74 | **97.26** | **96.96** |

*Table 3.* **AP on partially-inpainted fake images.** *Corvi* and *Rajan* struggle to detect partially-inpainted fake images, as they focus on real image features. Our detectors overcome this limitation.

| METHOD | 50% | 100% | 150% |
|---|---|---|---|
| CORVI | 6.13 | 83.36 | 98.05 |
| CORVI⊕ (OURS) | 97.48 | 99.92 | 99.98 |
| RAJAN | 49.69 | 99.93 | 100 |
| RAJAN⊕ (OURS) | **99.66** | **99.99** | **100** |

the final layer. We calculate the average precision (AP) and report the values in Table 4.

We evaluate these ablated models under the same setting described in Section 5.3. Our results, similar to those in Table 2, show that clamping without retraining leads to sub-optimal performance, likely due to improper reweighting of fake features. Training the entire backbone while clamping the final layer underperforms on FLUX images, likely due to the emergence of newly learned spurious fake features as discussed in Appendix A.6.

### 5.6. Robustness to other Post-Processing Artifacts

We also conduct tests testing the senstitivty of our detector to post-processing operations. We use JPEG Compression, additive gaussian noise and low-pass filtering to account for some of the common post-processing operations. We report our results in Appendix A.3.

## 6. Limitations

Our approach enhances existing detectors by ensuring that the final layer ignores features associated with the real distribution. However, patterns linked to the fake distribution can also be spurious. For instance, Rajan et al. (2025) observed that *Corvi* incorrectly associates upsampled images with the

fake distribution. We find that *Corvi⊕ (Ours)* exhibits a similar issue. To illustrate this, we take 500 Redcaps images (512×512), upsample them, and analyze how their upsampled versions affect the logit score. As shown in Fig. 7, we can see that upsampled images are more likely to be classified as fake. Despite the improvements we demonstrate, we emphasize that greater care must be taken when curating the set of real and fake images to avoid such behavior.

Our method encourages the detector to ignore features that are specific to real images. However, after the first stage of training, these features can still implicitly influence the detector's understanding of what constitutes a fake. For example, the model may learn that the absence of certain real-specific cues is indicative of a fake image, effectively using real image features in a negative sense. We observe such behavior in our own detector. To illustrate this effect, we include a simple toy example in Appendix A.6. These limitations suggest that better generalization could be achieved by extending this approach to train the entire network, as opposed to just the final layer.

## 7. Related Work

Training-based methods create effective fake image detectors. Wang et al. (2020) demonstrated that data augmentations during training improve generalization. Odena et al. (2016) demonstrated identifiable artifacts, like checkerboard patterns, in the Fourier transforms of fake images. Building on this, Zhang et al. (2019) trained on Fourier images, leading to improvements. Chai et al. (2020) improved detection using patch-based classification. Gragnaniello et al. (2021) removed downsampling in the initial layers and applied patch-wise training for further gains. These techniques were extended to the LDM setting by Corvi et al. (2023). Additionally, works such as Chen et al. (2024a) and Rajan et al. (2025) also use reconstructions of real images as part of the training data. These approaches struggle to general-

*Table 4.* **AP for different ablations.** We experiment with different ways of applying the stay-positive algorithm. We observe that in both settings, freezing the backbone of the network while re-training the last-layer performs the best. AVG is the equally weighted AP average across all generators.

| Method | SD | MJ | KD | PG | PixArt | LCM | FLUX | Wuerstchen | aMUSEd | AVG |
|---|---|---|---|---|---|---|---|---|---|---|
| Corvi | 97.87 | 94.81 | 95.38 | 91.40 | 94.46 | 96.33 | 74.57 | 95.83 | 94.57 | 92.80 |
| Corvi + clamped (no retrain) | 85.75 | 76.44 | 85.41 | 82.28 | 93.21 | 85.12 | 63.60 | 85.71 | 66.70 | 80.47 |
| Corvi + clamped (retrain) | 98.93 | 95.38 | 95.42 | 97.64 | 94.98 | 95.54 | 69.50 | 96.76 | 95.80 | 93.33 |
| Corvi⊕ (Ours) | 98.94 | 94.92 | 97.71 | 97.87 | 98.59 | 98.73 | **94.23** | **98.16** | 95.47 | 97.18 |
| Rajan | **99.40** | **98.30** | 98.18 | **98.62** | 98.64 | **99.79** | 87.80 | 94.51 | 95.38 | 96.73 |
| Rajan + clamped (no retrain) | 95.01 | 90.83 | 92.72 | 94.57 | 98.87 | 96.50 | 70.47 | 81.71 | 72.85 | 88.17 |
| Rajan + clamped (retrain) | 99.19 | 96.30 | 95.49 | 98.66 | 95.93 | 96.95 | 58.68 | 88.73 | 94.66 | 91.62 |
| Rajan⊕ (Ours) | 99.22 | 96.98 | **98.22** | 98.53 | **99.11** | 99.57 | 91.85 | 94.74 | **97.26** | **97.28** |

ize across architectures. To address this, Ojha et al. (2023) proposed using general-purpose visual encoders like CLIP (Radford et al., 2021) for fake image detection. Unlike ours, these methods explicitly rely on real features.

Contrary to prior approaches, GenDet (Zhu et al., 2023) treats fake image detection as an anomaly detection problem, similar to ours, but focuses on learning the real image distribution. Another paradigm suggests generators reconstruct fake images more easily than real ones. For reconstruction, Pasquini et al. (2023) use GAN inversion (Xia et al., 2022), DIRE (Wang et al., 2023), ZeroFake (Sha et al., 2024) apply DDIM inversion (Song et al., 2021), and AEROBLADE (Ricker et al., 2024) leverages a latent diffusion autoencoder. However, these methods at times struggle with detecting post-processed images (Rajan et al., 2025). A contrasting approach, akin to image reconstruction, was recently proposed by Cozzolino et al. (2025), who use neural image-compression networks (Cao et al., 2020) to model real distribution likelihood, assuming fake images are less likely to be part of it. While this method models the real distribution, we argue the focus should be on detecting fake image artifacts instead.

In the literature on mitigating spurious correlations, two-stage training strategies have been previously proposed. Liu et al. (2021) select samples with high training loss in the first stage and re-train the network on these samples in the second stage. Kirichenko et al. (2023) employ a last-layer re-training in a strategy. However, while these approaches rely on data curation in the second stage to force the network to focus on relevant features, our approach instead constrains the network architecture itself, encouraging it to focus on patterns that are present in fake images.

## 8. Conclusion

We showed that learning patterns from the real distribution can harm fake image detection and propose a last-layer fine-tuning strategy to improve performance. By ignoring

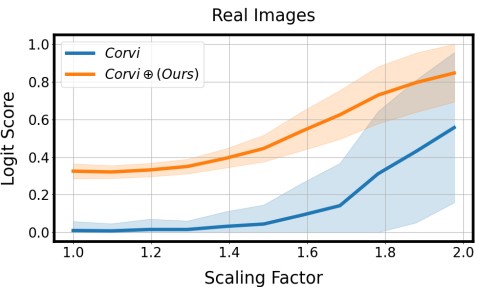

*Figure 7.* **Vulnerability to Spurious Fake Features.** Our method *Corvi⊕* is not able to mitigate spurious correlations pertaining to the fake distribution, where just like the original *Corvi*, it continues to associate upsampled images with the fake distribution.

real-distribution features, the model reduces susceptibility to spurious correlations, enhancing robustness. Additionally, models trained this way excel on partially inpainted real images. While this work focuses on detecting AI-generated images, we believe that our core idea — that specific patterns should not be associated with the real distribution could be applicable to other forms of media forensics, such as audio and video. We leave these directions to future research. We hope our study contributes to developing more robust detectors with positive societal impact.

## Acknowledgements

This work was supported in part by NSF IIS2404180, American Family Insurance, and Institute of Information & communications Technology Planning & Evaluation(IITP) grants funded by the Korea government(MSIT) (No. 2022-0-00871, Development of AI Autonomy and Knowledge Enhancement for AI Agent Collaboration) and (No. RS2022-00187238, Development of Large Korean Language Model Technology for Efficient Pre-training).

## Impact Statement

This paper aims to advance the field of fake image detection, with a focus on security and potential societal benefits, such as curbing misinformation. However, it may also provide bad actors with insights into recent developments in image forensics and expose vulnerabilities in existing systems.

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

# A. Appendix

## A.1. Implementation Details

We follow the training recipe used by Corvi et al. (2023). We train on 96 x 96 crops of the whole image using a batch size of 128. The data augmentations include random JPG compression and blur from the pipeline proposed by Wang et al. (2020). Following Gragnaniello et al. (2021), grayscale, cutout and random noise are also used as augmentations. Finally, in order to make the network invariant towards resizing, the random resized crop was added. For the baseline detectors as well as our re-trained variant, we report the average across two trained networks. When applying Algorithm 1, we use a batch size of 1024 to perform last-layer retraining. The rest of the training recipe does not differ from the original model. The second stage training on average converges in about 15 epochs, which takes an additional 4 hours.

We use the validation set provided by Corvi et al. (2023) for our training. Just like our training set, the real images come from COCO/LSUN and the fake images are generated at 256 x 256 using LDM. During training, if the validation accuracy does not improve by 0.1% in 10 epochs the learning rate is dropped by 10x. The training is terminated at learning rate $10^{-6}$. We adopt this following Wang et al. (2020).

During inference, we do not to crop/resize the image to a fixed resolution. This is possible since the ResNet-50 uses a Spatially-Adaptive Average Pooling layer before inference.

## A.2. Performance on Real Images

When testing our models effectiveness in detecting images coming from various kinds of generators, we use a real image dataset which consists of both natural real images (Redcaps), artistic real images (wikiart and LAION-Aesthetics) as well as face images (whichfaceisreal). We also post-process the images to simulate a real world setting. However, in this section, we test whether this set of real images is truly representative of the multiple real distributions present. In order to verify this, we plot the distributions of various kinds of real images.

### A.2.1. DATASET

**i. Test Real**: 6,000 images previously used in the main paper, containing artistic and natural images. **ii. GTA**: 6,382 GTA landscape images from the IMLE dataset (Li et al., 2019), providing a diverse range of synthetic scenes. **iii. ImageNet**: 8,000 real images from the GenImage benchmark, originally part of the ImageNet dataset, to capture standard real-world content. **iv. Cubism**: 2,235 images in the Cubism style sourced from the WikiArt dataset, adding a distinct artistic domain. **v. Pop Art**: 1,483 images in the Pop Art style from WikiArt, expanding the artistic domain with vibrant and modern aesthetics. **vi. Modern Art**: 4,334 images of Modern Art from WikiArt, offering a rich and varied artistic representation.

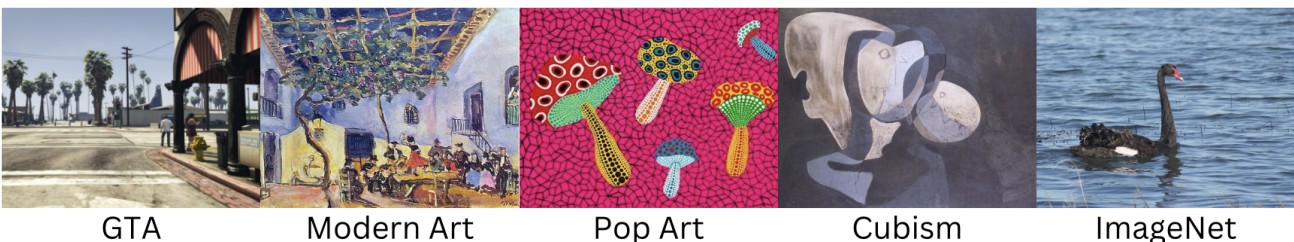

*Figure 8.* Example of different kinds of real images that we consider.

This dataset ensures a wide range of testing scenarios, from standard real-world distributions to highly diverse artistic and CG (but not neural network generated) domains. An example of the images used can be found in Figure 8.

### A.2.2. RESULTS

We pass the real images through *Corvi⊕ (Ours)* and *Rajan⊕ (Ours)* and plot the logit score (output of the network) in the form of violin plots in Fig 9. We observe that the fakeness scores of our test distribution indicated by "Test Real" is extremely similar to the other distributions of real images such as GTA, Cubism etc. This shows that the test-set which we

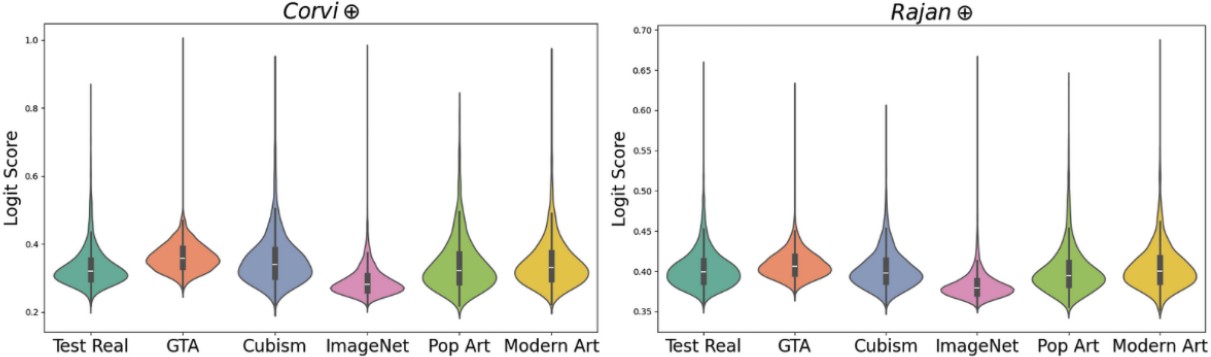

*Figure 9.* **Distribution of different kinds of real images.** The violin plots of *Corvi*⊕ (left) and *Rajan*⊕ (right) show that the test set used in our experiments accounts for a wide variety of real image types.

use in the main paper is representative of various different types of real image families. It is important to note that these real images are vastly different, belonging to different domains such as realistic scenes, art styles and video game rendered environments. However, the thing in common is the absence of the generator artifacts which is indicated by the low logit score in these violin plots.

### A.3. Robustness Analysis

In the main paper, we show that our method shows improved robustness to WEBP Compression and downsizing. In this section, we test the robustness of our detector to JPEG Compression, additive gaussian noise and low-pass filtering. For JPEG and additive noise, we follow the same experimental setting adopted by Rajan et al. (2025). Instead of AP, we report the logit scores. This way, we study the behaviour of the detector on both post-processed real and fake images,

We report the results from robustness tests in Fig 10. We can observe that our variants perform similar to the original models, which is not surprising since all of these perturbations were seen during training. Additionally, our real distribution shows very little variance unlike the distribution of the original baselines since our method does not rely on specific features to call an image real. Additionally, we also test the robustness to resizing using the SynthBuster dataset (akin to Fig 8 from Cozzolino et al. (2024)) and we report our results in Fig 11.

### A.4. Performance on GenImage

#### A.4.1. EXPERIMENTAL SETUP

In this section, we evaluate the performance of various detectors on the GenImage benchmark (Zhu et al., 2024). We consider the same baselines which we use in Section 5.3. The scope of our work is detecting images from the same kind of generative model (latent diffusion models), therefore we consider the GenImage testsets from Midjourney (mid), VQDM (Gu et al., 2022), SDv1.4, SDv1.5 (Rombach et al., 2022), Wukong (MindSpore).

GenImage has a set of real images and a set of fake images for each category, to ensure uniformity, we report the accuracy. However, the real images are always sourced from ImageNet, therefore, we also report the AP with respect to the corresponding real distribution. For accuracy, we use a common threshold of 0.5.

#### A.4.2. RESULTS

In Table 5 we report the accuracy (left) and AP (right). Based on the AP values, we can conclude that the methods which finetune the whole network represented by *DRCT*, *Corvi*, *Rajan*, along with our improved detectors perform better than the CLIP-based techniques represented by ClipDet and the original UFD methods. Furthermore, the AP in detecting VQDM generated images, improves for *Rajan*⊕ *(Ours)* by 4.15%. VQDM is a latent diffusion model which has a

*Table 5.* **Accuracy and AP of existing detectors on latent diffusion images from the GenImage benchmark.** Accuracy is computed with a fixed threshold, while AP is computed with respect to the real distribution (ImageNet). Whole network training-based methods (*DRCT, Corvi, Rajan, Corvi⊕, Rajan⊕*) outperform CLIP-based methods (*UFD, ClipDet*). On VQDM-generated images, *Corvi⊕ (Ours)* and *Rajan⊕ (Ours)* demonstrate improvements over their original counterparts. Additionally, on settings where *Corvi, Rajan* already perform well, our method performs the same if not slightly better.

| Accuracy (%) | MJ | SD1.4 | SD1.5 | Wukong | VQDM | AP (%) | MJ | SD1.4 | SD1.5 | Wukong | VQDM |
|---|---|---|---|---|---|---|---|---|---|---|---|
| UFD-ProGAN | 68.17 | 78.60 | 78.60 | 81.05 | 81.34 | UFD-ProGAN | 74.46 | 89.19 | 88.91 | 93.03 | 95.52 |
| UFD-LDM | 56.24 | 63.75 | 63.56 | 71.05 | 85.43 | UFD-LDM | 74.61 | 86.56 | 86.19 | 91.34 | 96.65 |
| Cozzolino-LDM | 67.91 | 85.32 | 85.71 | 78.40 | 82.70 | Cozzolino-LDM | 87.28 | 96.31 | 96.38 | 93.76 | 95.78 |
| DRCT-ConvNext | 94.43 | 99.37 | 99.19 | 99.25 | 76.84 | DRCT-ConvNext | 99.39 | **99.99** | **99.98** | **99.99** | 96.71 |
| Corvi | **99.67** | **99.99** | **99.89** | **99.97** | 80.86 | Corvi | **99.99** | 99.99 | **99.98** | **99.99** | 98.68 |
| Corvi⊕ (Ours) | 99.31 | 99.76 | 99.63 | 99.73 | **98.97** | Corvi⊕ (Ours) | 99.94 | **99.99** | 99.96 | **99.99** | **99.94** |
| Rajan | 96.58 | 99.88 | 99.85 | 99.87 | 74.67 | Rajan | 99.54 | **99.99** | 99.94 | **99.99** | 95.11 |
| Rajan⊕ (Ours) | 98.68 | 99.83 | 99.79 | 99.79 | 90.02 | Rajan⊕ (Ours) | 99.91 | **99.99** | 99.96 | **99.99** | 99.26 |

*Table 6.* **UFD Benchmark AP Scores across various diffusion models and detectors.** Detectors trained with our improved algorithm (*Corvi⊕, Rajan⊕*) consistently outperform their base versions and other models across all UFD benchmark settings, demonstrating superior generalization and robustness.

| | GLIDE | | | ADM | DALLE | LDM | | | AVG |
| Method | 100-27 | 100-10 | 50-27 | | | 100 | 200 | 200-cfg | |
|---|---|---|---|---|---|---|---|---|---|
| Corvi | 79.28 | 86.52 | 83.95 | 44.51 | 97.15 | 99.90 | **100.00** | 99.99 | 86.41 |
| Corvi⊕ (Ours) | **88.61** | **91.17** | **89.24** | **74.52** | **99.50** | **100.00** | **100.00** | 99.99 | **92.88** |
| Rajan | 71.51 | 77.57 | 77.54 | 44.63 | 89.19 | 99.90 | **100.00** | 99.99 | 82.54 |
| Rajan⊕ (Ours) | **80.43** | **83.50** | **83.36** | **62.32** | **98.63** | **100.00** | **100.00** | 99.99 | **88.53** |

different architecture compared to LDM. This improvement further demonstrates the effect of ignoring real image artifacts. Furthermore, the accuracy measurements demonstrate that without calibrating the threshold, our detectors are able to detect latent diffusion generated images in the GenImage benchmark.

## A.5. Performance on the UFD Benchmark

### A.5.1. Experimental Setup

In this section, we evaluate *Corvi ⊕ (Ours)* and *Rajan ⊕ (Ours)* on pixel-space diffusion models such as ADM (Dhariwal & Nichol, 2021) and GLIDE (Nichol et al., 2022), as well as on autoregressive models such as DALL-E (Ramesh et al., 2021) and Stable Diffusion (Rombach et al., 2022). Since our focus is on comprehensive detection of images from a known generator family, we do not consider GAN-generated images in this evaluation.

### A.5.2. Results

We report the results in Table 6. Both *Corvi⊕* and *Rajan⊕* demonstrate substantial improvements over their original counterparts in terms of AP on images generated by GLIDE, ADM, and DALL-E. These results indicate that, even when generalizing to entirely unseen settings, disregarding real-image features remains an effective strategy.

## A.6. Spurious Correlations in the Real Distribution Could Influence Fake Features

In our work, we demonstrated that features associated with the real distribution harm the detectors performance, and proposed a way to mitigate such an issue by ignoring the real features under certain assumptions. However, our method relies on re-training the final layer of the neural network. We demonstrate a way in which such a detector can still suffer from the effects of features associated with the real distribution. We rely on a toy example, hypothesizing such a case.

### A.6.1. A Toy Example

Let us consider the dataset used to train the detector by Rajan et al. (2025). The training data contains real images from COCO and LSUN. The LSUN images are WEBP compressed. The precise details can be found in Section 3. In the main paper, we demonstrate that the existing detector can associate the presence of WEBP compression artifacts with real images. However, we hypothesize that the detector can actually also associate the absence of WEBP compression artifacts with fake images.

To explain our intuition, we consider a simple feedforward neural network, with ReLU activations. This network has been trained using the dataset described above. We illustrate the network in Fig 12.

**WEBP Artifact Sensitivity**  Neural networks, such as ResNet-50, exhibit sensitivity to WEBP compression artifacts, suggesting the presence of a neuron that selectively activates in response to images containing such artifacts. Given a ReLU activation function, this neuron outputs a high activation magnitude for WEBP-compressed images from the real distribution, particularly LSUN images, while producing a low or zero activation magnitude for images without WEBP compression, such as COCO images and those from the fake distribution.

**Flipping the Activation to Detect Absence of WEBP**  If this activation is multiplied by a negative weight, followed by the addition of an appropriate bias and the application of a ReLU activation, the network can learn a transformed version of this neuron that activates only when WEBP artifacts are absent. This means the network could, in principle, learn to detect images that have *not* undergone WEBP compression by repurposing the same underlying feature.

**Combining WEBP and LDM Artifacts to Learn an AND Condition**  Beyond compression artifacts, the network also exhibits sensitivity to generative model artifacts, such as those present in latent diffusion model (LDM)-generated images. A similar mechanism can be hypothesized for LDM artifacts, where a neuron activates in response to their presence. During training, for a fake image, both the absence of WEBP artifacts and the presence of LDM artifacts hold. If the network learns to associate this specific combination with the fake class, it effectively learns an *AND* condition—where a fake image is recognized only when WEBP artifacts are absent and LDM artifacts are present.

**Impact on Detector Robustness**  If the detector associates both the presence of LDM-artifacts *AND* the absence of WEBP artifacts with fake images, applying WEBP compression to a fake image during inference could contradict this learned *AND* condition. The addition of WEBP artifacts could weaken the distinguishing signal, reducing the activation of the fake-detecting neuron and thereby lowering the fake score. In fact, such a situation does arise in our experiments.

To illustrate this we revisit some of our previous experiments, in Fig. 1 from Section 3.1, we show that excluding WEBP-compressed real images during training can yield strong detector performance, as seen in the method labeled *Rajan (only COCO)*. This suggests that removing WEBP-related biases from the training data allows the model to generalize better, avoiding reliance on compression artifacts as a shortcut for classification.

In Fig. 4, our method shows improvements over the baseline, demonstrating that explicitly mitigating spurious correlations can enhance detector performance. However, despite this improvement, the detector is still not as robust as *Rajan (only COCO)*. This can be explained by our earlier toy example: if the model has learned an implicit rule where the absence of WEBP artifacts is a necessary condition for an image to be classified as fake, then introducing WEBP compression to fake images weakens the fake classification signal. Consequently, while removing WEBP-compressed real images from training reduces bias, it does not fully eliminate the underlying vulnerability, since the model may still rely on other spurious correlations to distinguish real from fake.

### A.7. Improved Detection of GAN-generated Images

Our work in the main paper focused on improving the detection of LDM (Rombach et al., 2022) generated images. In this section, we extend the results to the detection of images generated by GANs (Goodfellow et al., 2014). We consider a baseline ResNet-50 trained using the dataset created by Wang et al. (2020). This dataset consists of 360k real images taken from LSUN and 360k images generated by ProGAN (Karras et al., 2018). We use the same training recipe described in Appendix A.1, however, we do not employ the random resized crop data augmentation in this case. We train a ResNet-50 on this dataset. We refer to this baseline as *GAN-Baseline*, we re-train the last layer of this detector to ignore real features. We refer to this model as *GAN-Baseline⊕ (Ours)*.

*Table 7.* **Accuracies of various detectors on CNN-generated images.** In most settings, our method *GAN-Baseline⊕*, outperforms/maintains performance with respect to the original detector. Additionally, on most settings where *GAN-Baseline* already perform well, our method performs the same if not slightly better.

| METHOD | PROGAN | STYLEGAN | STYLEGAN2 | GAUGAN | BIGGAN | CYCLEGAN | STARGAN | DEEPFAKE | SAN | CRN | IMLE | WFCIR | AVG |
|---|---|---|---|---|---|---|---|---|---|---|---|---|---|
| UFD-PROGAN | 99.81 | 84.92 | 74.96 | **99.47** | 95.07 | **98.33** | 95.74 | **68.56** | 56.62 | 56.58 | 69.10 | 87.20 | 82.20 |
| UFD-LDM | 93.18 | 83.20 | 83.95 | 90.75 | 87.90 | 94.62 | 84.36 | 54.46 | **79.22** | 74.33 | 85.00 | 70.35 | 81.78 |
| CLIPDET | 72.96 | 70.49 | 70.86 | 83.94 | 74.10 | 87.28 | 54.95 | 50.87 | 77.62 | 53.10 | 53.69 | 53.10 | 66.91 |
| GAN-BASELINE | **100.00** | 99.52 | 93.34 | 95.53 | 96.65 | 93.14 | 99.54 | 55.43 | 54.33 | **99.96** | **99.87** | **100.00** | 90.61 |
| GAN-BASELINE⊕ | **100.00** | **99.82** | **98.96** | 94.52 | 95.02 | **93.30** | **99.77** | 68.17 | 75.57 | 86.75 | 86.75 | 99.70 | **91.53** |

*Table 8.* **AP of various detectors on CNN-generated images.** In most settings, our method *GAN-Baseline⊕*, outperforms/maintains performance with respect to the original detector. Additionally, on most settings where *GAN-Baseline* already perform well, our method performs the same if not slightly better.

| METHOD | PROGAN | STYLEGAN | STYLEGAN2 | GAUGAN | BIGGAN | CYCLEGAN | STARGAN | DEEPFAKE | SAN | CRN | IMLE | WFCIR |
|---|---|---|---|---|---|---|---|---|---|---|---|---|
| UFD-PROGAN | 99.99 | 97.56 | 97.89 | **99.98** | 99.26 | **99.79** | 99.37 | 81.76 | 78.80 | 96.58 | 98.60 | 97.27 |
| UFD-LDM | 99.68 | 93.36 | 92.56 | 99.78 | 98.04 | 99.83 | 97.51 | 70.52 | 90.40 | 83.96 | 93.88 | 96.77 |
| CLIPDET | 91.29 | 78.13 | 76.47 | 90.19 | 80.04 | 89.51 | 81.07 | 65.98 | 85.65 | 64.63 | 55.66 | 64.63 |
| GAN-BASELINE | **100.00** | **99.99** | **99.99** | 99.78 | **99.74** | 98.73 | 99.98 | 94.27 | 88.79 | **99.99** | **99.99** | **100.00** |
| GAN-BASELINE⊕ | **100.00** | **99.99** | **99.99** | 99.30 | 99.03 | 98.93 | **99.99** | 95.45 | **94.46** | **99.99** | **99.99** | 99.99 |

### A.7.1. IMPROVED ROBUSTNESS TO WEBP COMPRESSION

Our dataset uses real images from LSUN. Consequently, our baseline detector (*GAN-Baseline*) also associates WEBP compression artifacts with the real distribution, similar to the observations recorded in Section 3.1. We conduct an experiment to compare our retrained detector *GAN-Baseline⊕ (Ours)* with the original detector.

**Experiment Details**: With this experiment, we intend to measure the sensitivity of the original *GAN-Baseline* to WEBP compression and measure if our method can mitigate these issues. We use the StyleGAN (Karras et al., 2019) test set provided by Wang et al. (2020). We sample 500 real images and 500 fake images randomly. We apply different levels of WEBP compression to the fake images and compute the AP, analogous to our measurement in Section 3.1.

**Analysis**: We present the results in Fig. 13. WEBP compression significantly degrades the performance of the baseline detector. In contrast, our detector demonstrates enhanced robustness to WEBP compression. This provides further evidence suggesting that avoiding reliance on real features can improve robustness against such spurious correlations.

### A.7.2. GENERAL PERFORMANCE

In this section, we study the generalization ability of the detector in comparison to other baseline detectors. We use the dataset consisting of CNN generated images from Wang et al. (2020). The dataset consists of ProGAN (Karras et al., 2018), StyleGAN (Karras et al., 2019), StyleGAN2 (Karras et al., 2020), GauGAN (Park et al., 2019), BigGAN (Brock et al., 2019), CycleGAN (Zhu et al., 2017), StarGAN (Choi et al., 2018) as well as Deepfakes (Rossler et al., 2019), super-resolution (SAN) (Dai et al., 2019) as well as networks which use a perceptual loss to refine images, such as CRN (Chen & Koltun, 2017) and IMLE (Li et al., 2019). It also consists of high quality GAN-generated faces in WFIR (whi). This test set comes with both real and fake images for each generator category. Therefore, we report both accuracy with a 0.5 threshold and the AP.

We report the accuracy and AP scores in Tables 7 and 8, respectively. Consistent with our findings in Table 2, we observe that the full-network training paradigm represented by *GAN-Baseline* and *GAN-Baseline ⊕ (Ours)* outperforms the CLIP-linear probing paradigm represented by the UFD methods and ClipDet. Our method, *GAN-Baseline⊕*, outperforms the *GAN-Baseline* detector by 5.67 AP on images generated through super-resolution. Since super-resolution operates on real images, we hypothesize that the real features in these images harm the performance of the *GAN-Baseline* detector. However, our method is able to better detect such images since it does not rely on real features. Interestingly, our method shows lower accuracy on the CRN and IMLE test sets. Upon closer inspection, we find that the real distribution in these test sets consists of video game images from the GTA series, which differ significantly from the natural distribution. Notably, some patterns that the detector associates with fake images are also present in these video game images. To verify this, we conduct an experiment similar to the one in Section A.2. Except for the class "Test Real", all other classes use the same images as

described in Section A.2. We sample the 'Test Real' distribution from the real images in the StyleGAN test set. We observe that most of the real images, except for the GTA images, exhibit similar logit scores. This shows that our model still does a good job in generalizing to different domains of real images, however, the performance on some unrelated domains can drop.

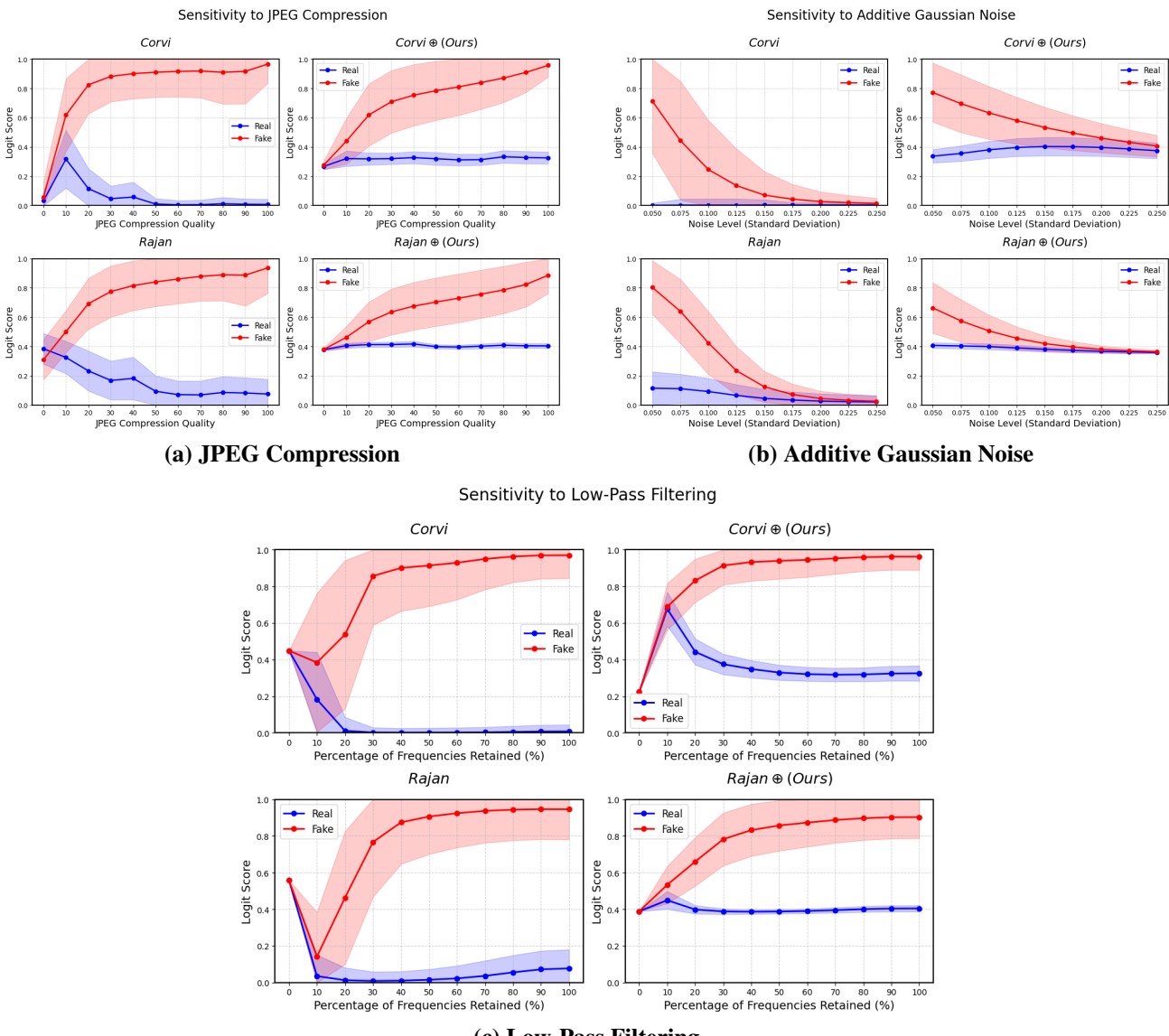

(a) **JPEG Compression**

(b) **Additive Gaussian Noise**

(c) **Low-Pass Filtering**

*Figure 10.* **Robustness to Common Post-Processing Artifacts.** We observe similar trends across various corruptions (JPEG compression, additive noise, and low-pass filtering) between the original detectors and our improved versions. Note that these perturbations were part of the training data augmentations. More importantly, the real image distribution exhibits very uniform logit scores, since our approach relies on the absence of generator artifacts to identify these images.

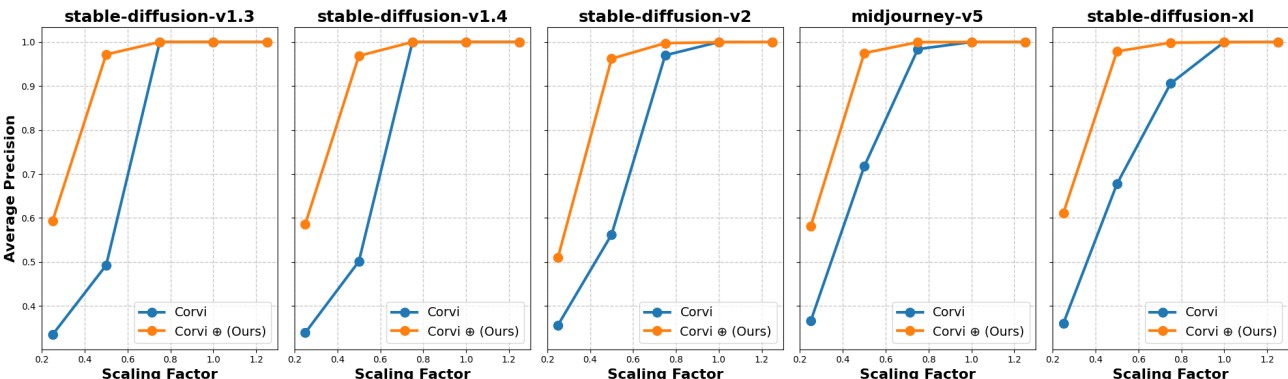

*Figure 11.* **Robustness to Resizing (SynthBuster)** We can observe that our detector, *Corvi⊕* shows improved robustness to downsizing compared to the original *Corvi*

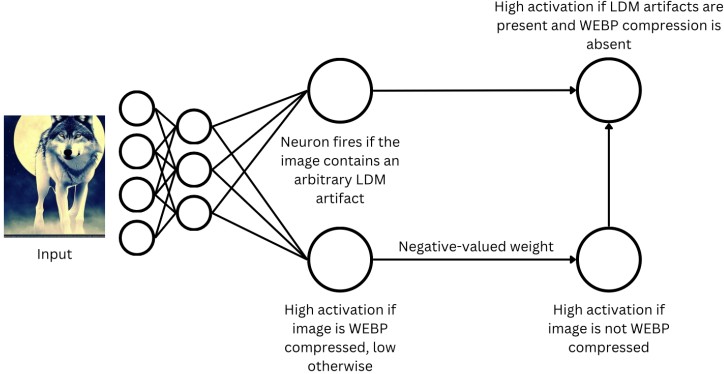

*Figure 12.* **Spurious Fake Features.** In this neural network, the circles demonstrate neurons, some neurons are bigger in size for demonstration purpose. For such a case, the detector can associate the absence of a spurious real image artifact, such as WEBP compression with fake images.

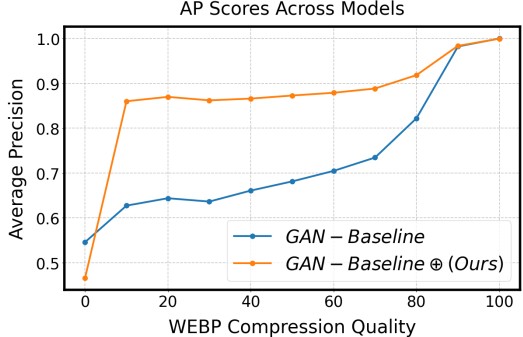

*Figure 13.* **Improved Robustness to WEBP compression (GAN case).** Compared to the original GAN-Baseline, our model displays an improved robustness to WEBP compression.

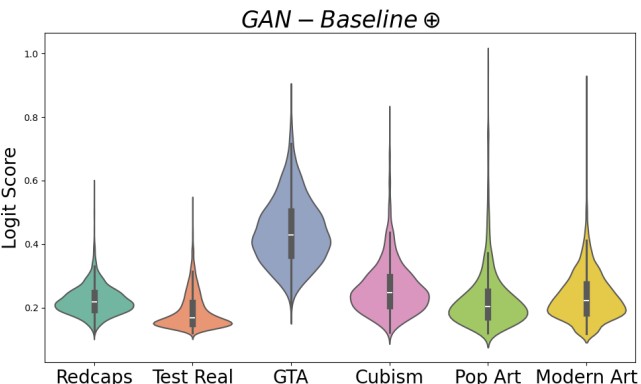

*Figure 14.* **Distribution of different kinds of real images.** We observe that most types of real images are assigned a similar value of fakeness. An exception is the GTA-based images which has a relatively higher score indicating the presence of spurious fake features.

