# OpenReview forum: "Stay-Positive: A Case for Ignoring Real Image Features in Fake Image Detection"
_ICML.cc/2025/Conference — ICML 2025 poster_

### Official Review · Reviewer_Qhvq · 2025-03-03

**Overall Recommendation:** 3

**Summary:**

The paper addresses AI-generated image detection challenges, showing that detectors rely on spurious real-image artifacts. It introduces Stay-Positive, which retrains the last layer to focus only on fake features by enforcing non-negative weights.
Key contributions:
1.   	Identifies spurious correlations (e.g., WEBP compression) that mislead detectors.
2.   	Stay-Positive algorithm improves generalization by removing reliance on real-image artifacts.
3.   	Enhances robustness to post-processing, resizing, and newer models (FLUX, aMUSEd).
4.   	Improves detection of partially AI-modified images, aiding forensic applications.
The findings suggest ignoring real-image features improves fake detection and reduces misclassification risks.

**Claims And Evidence:**

The paper provides strong empirical support for its main claims. It evaluates multiple generative models (LDM, FLUX, aMUSEd, GANs) and tests post-processing artifacts, resizing, and inpainting, making the results robust.
Well-Supported Claims:
·       Spurious Correlations in Detection: The study shows real-image artifacts (e.g., WEBP compression, downsizing) cause misclassification, supported by fake and real score distributions.
·       Stay-Positive Improves Robustness: The method re-trains only the last layer with non-negative weights, preventing reliance on real-image features. Tables 1 and 2 confirm its effectiveness.
Claims Needing More Support:
·       Post-Processing Artifacts: The study focuses mainly on WEBP compression. Testing JPEG, PNG, and frequency-based artifacts is needed to rule out dataset bias.
·       FLUX Misclassification due to real features: FLUX is out-of-distribution for the model. Testing diverse generative models (e.g., DALL·E, Imagen) would clarify if this is a broader limitation.

**Essential References Not Discussed:**

The paper compares Stay-Positive to state-of-the-art zero-shot approaches like CLIP-based detectors (CLIPDet) and universal fake image detectors (UFD).
·       Findings: While CLIP-based zero-shot detectors perform well on seen distributions, they struggle with new diffusion models (e.g., FLUX, aMUSEd), whereas Stay-Positive improves generalization.

There are not many cities of formal statistical tests, and zero-shots works such as:
·       Manifold induced biases for zero-shot and few-shot detection of generated image.
·       ZeroFake: Zero-Shot Detection of Fake Images Generated and Edited by Text-to-Image Generation Model
Which aligns with the idea of focusing on fake features generated by LDMs.

**Experimental Designs Or Analyses:**

Yes, I reviewed the experimental setups and analyses.
 Below are the key experiments and their validity:
1.   	Case Study 1: Post-Processing Artifacts
a.       Post-Processing Artifacts: The experiment confirms that WEBP compression causes misclassification due to spurious correlations. The results are valid.
b.       Issue: The study only tests WEBP; broader validation with JPEG, PNG, and other artifacts is needed.
c.      The resizing-based artifact analysis only uses images generated by SDv2.1, despite an existing benchmark dataset for this type of evaluation. the study should leverage an existing benchmark – for example, https://github.com/grip-unina/ClipBased-SyntheticImageDetection/tree/main


2.       Case Study 2: Generalization Across Generators
a.      The experiment examines how Corvi-trained detectors struggle with FLUX-generated images, suggesting reliance on spurious real-image features.
b.       The analysis is well-supported by fake and real score distributions, which show FLUX images receive higher real scores than LDM ones.
c.       Issue: FLUX images are out-of-distribution for the model. More tests on other out-of-distribution generators (e.g., Imagen, DALL·E) could indicate whether this is a generalizable limitation.
3.       Stay-Positive Algorithm Validation
a.       Results confirm that the method reduces misclassification caused by spurious correlations, improving detection on post-processed and partially inpainted images.

**Methods And Evaluation Criteria:**

Yes, the model is trained on LSUN, COCO, Redcaps and evaluated on LDM, FLUX, aMUSEd, GANs. Which includes up-to-date models and comparison datasets. The Stay-Positive modification correctly re-trains only the final layer, enforcing non-negative weights by setting all negative values (which pushes prediction to zero, i.e. real) to zero.
This ensures detection relies only on fake features, improving robustness to post-processing (compression, resizing, inpainting).

**Other Comments Or Suggestions:**

Compare the proposed method with more zero-shot techniques to provide a broader evaluation.
2.      Further test the hypothesis from Case Study 2 by applying additional post-processing techniques and evaluating more generative models. This will strengthen the analysis derived from distribution plots, as the observed differences could also be attributed to a generalization gap.
3.      Some experiments were conducted on very small datasets, such as Section 5.1.2 (Resizing-based artifacts), which may impact the reliability of the findings.
4.      Certain tests utilized custom-generated images from Stable Diffusion 2.1 rather than pre-existing benchmark datasets (e.g., in Section 5.1.2). For instance, WhichFaceIsReal has a benchmark dataset based on COCO, and other open-source datasets are available as well. Using standardized datasets would improve reproducibility and comparability.

**Other Strengths And Weaknesses:**

Strengths:
The paper introduces a simple yet effective idea: classifying fake samples primarily based on fake features rather than relying on the real data distribution. This approach has several advantages:
a.       The second-stage fine-tuning is relatively fast to perform.
b.       It focuses on specific features and artifacts generated by different image generators.
c.       The method is practical and applicable to real-world scenarios.
d.       The empirical diagnostics are well-structured, intuitive, and logically sound.
e.      Robustness to inpainting detection.
Weaknesses:
The approach is supervised and trained on relatively small datasets, especially compared to zero-shot methods that leverage large-scale, pre-trained predictors.


Ignoring certain real-image features may lead to misclassification in cases where "fakeness" is absent in generated samples. This poses a generalization limitation, particularly when detecting more advanced or unseen generative models. Moreover, this approach may introduce a strong bias toward the fake distribution, potentially compromising the model's generalization.

**Questions For Authors:**

--

**Relation To Broader Scientific Literature:**

Unlike prior methods that focus on both real and fake features, this paper argues that fake image detection should be based exclusively on generative artifacts, disregarding any patterns related to real images, using a very simple method of second-stage training.
2.       Experiments show significant performance gains in detecting FLUX and aMUSEd-generated images, which were previously misclassified due to reliance on real-image artifacts.
3.       Prior detection methods fail to identify partially AI-modified images due to their reliance on real-image features.

**Theoretical Claims:**

The argument for using non-negative weights in the final layer appears well-founded within the framework of their proposed method. It effectively forces the model to focus exclusively on fake image features, thereby reducing reliance on spurious correlations from real data. Logically, this makes sense because a feature that contributes positively to the detection of fakeness is assigned a label of 1.
However, this technique introduces a bias toward the fake distribution, potentially limiting the model’s ability to generalize across different generators.

Spurious Fake Features Remain: While the method removes real-image-related biases, it does not necessarily address spurious correlations within fake features. The paper itself acknowledges this:
"Our approach ensures that the detector ignores real image-specific features, but these features can still shape its notion of fakeness."

---

> ### Author Rebuttal · Authors · 2025-04-01
>
> **Further test the hypothesis from Case Study 2 by applying additional post-processing techniques and evaluating more generative models. This will strengthen the analysis derived from distribution plots, as the observed differences could also be attributed to a generalization gap**
>
> **Test on other generators**: We would like to clarify that the generalization gap could manifest in two ways. First, the learned patterns associated with fake images may be absent in FLUX images. Second, while some signs of fakeness may be present in FLUX images, the decision may be influenced by spurious signs of realness. We believe the latter issue should be avoided, which our work demonstrates.
>
> The principle that relying on real features harms the detector holds true for generators beyond FLUX. We have evaluated our approach on aMUSEd (Tables 1, 2) and observed the same issue. This effect is also evident in VQDM (Table 4) and generators like DALL-E, GLIDE and ADM[1] (https://imgur.com/a/eLTN1Kv), reinforcing our claim that this is a broader limitation of existing methods, which our approach mitigates.
>
> **Post-Processing**:  We have tested our detector's sensitivity to JPEG compression, additive noise, and low-pass filtering. For the results please refer to the response to reviewer JDtd. However, we would like to note that PNG is a lossless compression format and does not introduce artifacts.
>
> We hope that this serves as evidence regarding the general nature of this limitation.
>
> **However, this technique introduces a bias toward the fake distribution, potentially limiting the model’s ability to generalize across different generators.**
>
> If an image lacks signs of fakeness, our method will not classify it as fake, which we do not consider a limitation. Given training images from a specific generator, we can only learn how that generator deviates from the real distribution. Without detectable artifacts, there is insufficient evidence to classify an image as fake. Our work is the first to show that when generalizing to images from a known family of generators (Introduction, lines 12–28), existing detectors underperform because they rely on real features.
>
> Without signs of “fakeness”, existing detectors may attempt to classify images as real based on learned patterns, but as shown in Sections 3 and 5.4, these features are mostly spurious, making the classification hypothesis unreliable despite potential success in some cases.
>
> Appendix A.2 shows that our detectors, Corvi+ and Rajan+, trained on COCO and LSUN real/fake images, perform similarly on real images from other domains, like GTA and artworks, despite these domains being unseen during training. This confirms that our method detects fakeness patterns, not just labeling out-of-distribution images as fake, reinforcing our core claim.
>
> **Spurious Fake Features Remain**
>
> It is true that spurious fake features can remain even after our method. However, we would like to clarify that the purpose of the study is to show that the real features learned by the neural network are spurious which end up harming the detector when it comes to generalization.
>
> **Use existing benchmarks for resizing studies**
>
> We thank the reviewer for sharing the Synthbuster benchmark for the resizing study. We have tested our approach on this benchmark, using a plot similar to Fig 8 from the Synthbuster paper.
>
> Results Link: https://imgur.com/a/SMYKgea
>
> The results show that our method effectively mitigates Corvi's spurious association of downsampling with realness, providing evidence of the generality of our algorithm.
>
> **Compare the proposed method with more zero-shot techniques to provide a broader evaluation.**
>
> We compare our detector with large model-based detectors like UFD and ClipDet, and observe that these methods do not perform as well as fully-trained neural networks, as shown in Table 2. We also tested the latent diffusion specific zero-shot detector AEROBLADE (Ricker et al., 2024), which fails to match the performance of our detectors. Therefore, we do not believe leveraging pre-trained techniques offers advantages over our approach.
> | AP  | SD     | MJ     | KD | PG | PixArt | LCM     | Flux    | Wuerstchen | aMUSEd  |
> |--------------------|--------|--------|-----------|------------|----------|---------|---------|------------|---------|
> | AEROBLADE | 90.81 | 96.48 | 94.03 | 71.53| 87.84|60.34| 88.39| 85.93| 88.39|
> | Corvi + (Ours)|98.94 | 94.92 | 97.71| 97.87| 98.59 | 98.74| **94.23**| **98.16**| 95.47|
> | Rajan + (Ours)| **99.23** | **96.98** | **98.22**| **98.53** | **99.11**| **99.57**| 91.85 | 94.74| **97.26** |
>
> **References not discussed**
>
> We would like to highlight the fact that we have discussed zero-shot methods (Related Work, line 417-428). However, we will also cite works such as ZeroFake in our final version.
>
> **References**
>
> [1] Dhariwal, P., & Nichol, A. (2021). Diffusion models beat gans on image synthesis. NeurIPS 21

---

> > ### Comment · Reviewer_Qhvq · 2025-04-06
> >
> > The authors addressed the main concerns with additional experiments and clarifications. They expanded their evaluation to include more generative models and post-processing methods, tested on the Synthbuster benchmark for resizing artifacts, and added comparisons with zero-shot detectors such as AEROBLADE. They indicated that these results will be included in the final version. These additions help support their claims and improve the completeness of the evaluation.
> >
> > The core idea is simple and practical. While the method is straightforward, it appears effective in reducing misclassification caused by spurious correlations and shows some robustness across different generative models and artifacts, though it still relies on supervised training and inherits the associated limitations.
> >
> > Given the strengthened empirical support and broader evaluation, I am increasing my score to 3.

---

### Official Review · Reviewer_GtjV · 2025-03-13

**Overall Recommendation:** 3

**Summary:**

This paper presents a method for improving fake image detection. It makes the observation that a fake image detector may learn spurious features associated with real images, such as post processing artifacts or image quality, which may lead to suboptimal detection performance. It thus proposes to constrain the detector to focus solely on artifacts that characterize fake images. Specifically, it focuses on the setting where the last layer of the fake image detector is a linear layer, where positive weights contribute to the likelihood of classifying as fake and negative weights reduce this likelihood. It proposes retraining this layer so that only positive weights contribute to the final prediction. Experiments show that this approach improves the robustness and consistency of fake image detectors.

## update after rebuttal

I am satisfied with the authors' rebuttal and keep my original score of 3.

**Claims And Evidence:**

The claims made in the paper are generally backed up by experimental analyses and/or citations of findings from existing works.

**Essential References Not Discussed:**

The paper sufficiently covers relevant literature in the field.

**Experimental Designs Or Analyses:**

The experimental designs and analyses are reasonable.

* The settings consider comprehensive scenarios, where the real images cover different domains and styles, and the fake images are sourced from various recent, widely used diffusion models and GAN models, with both full and partial (i.e. inpainting) generations. This covers realistic scenarios in practical applications and demonstrate generalizability of the proposed method. Results verify that the proposed method attains consistent performance improvements across these different conditions.

* Analyses are provided to support the claims that existing fake image detectors learn spurious correlations of real images, such as compression and downsizing, and show that the proposed method successfully mitigates the problem.

**Methods And Evaluation Criteria:**

[Method]

Strengths:

* The proposed method looks reasonable. Conceptually, it does address the motivation of preventing the spurious correlation associated with real images.

* It is lightweight and is compatible with other existing methods of learning fake images detectors.

Concerns:

* In Figure 7, it appears that the Redcaps images have much higher fakeness probabilities under the Corvi with the proposed method model than under the regular Corvi model, even at scaling factor 1. As scaling factor increases, the fakeness probability under Corvi+proposed method further increases -- e.g. under scaling factor 1.6, a Redcaps image on average gets fakeness probability above 0.5 under Corvi+proposed method, but only receives fakeness probability at around 0.1 on average under regular Corvi. Could the authors provide more discussion about this phenomenon?

* Related to the question above, since the proposed method retrains the last layer to focus on fake attributes, would it be a concern that spurious correlations associated with fake images might inadvertently be amplified through this process?

[Evaluation Criteria] The selection of benchmark datasets is reasonable and covers diverse real-world scenarios. The evaluation metrics are appropriate for this task.

**Other Comments Or Suggestions:**

[Minor Suggestions on Writing/Presentation]

* The second and third paragraphs of the introduction seem a bit repetitive and read like paraphrased versions of each other. They could be condensed into one.

* It could be helpful to add an “Average” column in the tables to make it easier to compare overall performance and consistency across different fake image sources among the baselines as well as the proposed method.

* Typo in Limitations - L415 should be "associates upsampled images with the *fake* distribution"?

**Other Strengths And Weaknesses:**

Please refer to other sections.

**Questions For Authors:**

* See the questions under Methods And Evaluation Criteria.

* Additionally, are there any insights or analyses of what specific types of real or fake features are learned by the detectors, e.g. by inspecting the weights, using saliency maps, or other interpretability methods, etc.?

**Relation To Broader Scientific Literature:**

This paper falls under the broader fields of data forensics and fake image detection. The paper is motivated by the observation that detectors trained with existing methods may inadvertently learn spurious correlations and the proposed method alleviates this problem and can be applied on top of various recent existing works.

**Theoretical Claims:**

N/A. This work does not involve theoretical claims or proofs.

---

> ### Author Rebuttal · Authors · 2025-04-01
>
> **In Figure 7, it appears that the Redcaps images have much higher fakeness probabilities under the Corvi with the proposed method model than under the regular Corvi model. Could the authors provide more discussion about this phenomenon?**
>
> We thank the reviewer for raising this point. In all our experiments (Sections 5.2, 5.3, and 5.4), we observe that Corvi+ performs better in detecting fake images, despite the apparent "higher probability of fakeness for real images" (further details in Appendix A.2). This issue arises from our use of the term "probability of fakeness," which we now recognize as a misleading interpretation.
>
> Our approach removes negative weights in the final layer, meaning it does not rely on "real features." As a result, the real score (Section 3.2, lines 156–159) is always zero, preventing our method from assigning highly negative scores for a given image. Consequently, for real images, the sigmoid output is closer to 0.5 rather than 0. This does not imply that a real image is half as likely to be fake, but rather represents a logit score that helps separate real and fake distributions. We will provide additional clarification for this in our final report.
>
> Both Corvi and Corvi+ learn the same spurious fake features, such as upsampling artifacts. However, because Corvi+ eliminates spurious correlations associated with the real distribution while retaining those linked to the fake distribution, it ultimately has fewer spurious correlations overall. This leads to better detection performance.
>
> **Related to the question above, since the proposed method retrains the last layer to focus on fake attributes, would it be a concern that spurious correlations associated with fake images might inadvertently be amplified through this process?**
>
> There is a possibility that spurious correlations associated with the fake distribution could be inadvertently amplified. However, the core goal of our work is to highlight that patterns associated with the real distribution are spurious and should not influence the decision. Ignoring these patterns improves the detector's performance, independent of the spurious features in the fake distribution.
>
> **Additionally, are there any insights or analyses of what specific types of real or fake features are learned by the detectors, e.g. by inspecting the weights, using saliency maps, or other interpretability methods, etc.?**
>
> We have identified and explained specific artifacts linked to each distribution. In Section 3, we show that compression artifacts can be associated with real images, and low-level artifacts may be spuriously linked to the real distribution due to quality differences in the training data (Section 3.2). Further evidence is provided in Section 5.3.2 (lines 372–384), where post-processing FLUX- and aMUSEd-generated images significantly improves the performance of the original Corvi detector (from 57.25 to 74.57 on FLUX) and Rajan detector (from 80.64 to 87.80 on FLUX), suggesting that post-processing removes spurious low-level features which the detector associates with the real distribution.
>
> We also experimented with GradCAM [1], but the activation maps did not provide clear insights, so we did not discuss them in our work.
>
> **Minor Suggestions**
>
> We thank the reviewer for providing us with these minor suggestions, we will incorporate these changes in the final version.
>
> **References**
> 1. Selvaraju, Ramprasaath R., et al. "Grad-cam: Visual explanations from deep networks via gradient-based localization." Proceedings of the IEEE international conference on computer vision. 2017.

---

> > ### Comment · Reviewer_GtjV · 2025-04-07
> >
> > Thank the authors for the rebuttal. I am content with the response and will keep my score.

---

### Official Review · Reviewer_JDtd · 2025-03-13

**Overall Recommendation:** 3

**Summary:**

This paper introduces "Stay-Positive," an algorithm designed to improve AI-generated image detection by focusing solely on generative artifacts while disregarding features associated with real images. The authors argue that spurious correlations, such as compression artifacts that detectors mistakenly associate with real data distribution, significantly impact detection performance. Their key insight is that an image should be classified as fake if and only if it contains artifacts introduced by the generative model.

The proposed method involves retraining the last layer of existing detectors to constrain them to focus exclusively on generative artifacts. Through extensive experimentation, the authors demonstrate that Stay-Positive improves detector performance in several ways: (1) reducing susceptibility to spurious correlations, (2) enhancing generalization to newer generative models within the same family, and (3) increasing robustness to post-processing operations like compression and downsizing. Notably, the authors show substantial improvements on challenging generators like FLUX and aMUSEd compared to baseline methods.

**Claims And Evidence:**

The claims made in the paper are generally well-supported by empirical evidence. The authors provide comprehensive experimental results that demonstrate:

- The effectiveness of Stay-Positive in improving detection performance across multiple generative models, with particularly significant gains for challenging models like FLUX (42.08 AP improvement) and aMUSEd (10.62 AP improvement).
- Enhanced robustness to post-processing operations like compression and downsizing, which is demonstrated through controlled experiments and visualization.
- The premise that focusing solely on fake image features rather than real image features yields better generalization.

These claims are supported by quantitative results presented in tables and figures, showing performance metrics (Average Precision) across different scenarios and comparing against established baselines. The experimental design includes appropriate controls and covers a wide range of generative models and post-processing techniques.

**Essential References Not Discussed:**

I think the references in the manuscript are relatively comprehensive.

**Ethics Expertise Needed:**

["Other expertise"]

**Experimental Designs Or Analyses:**

The experimental design appears sound and comprehensive. I examined the main experiments presented in the paper:

- Comparison with baseline detectors (Corvi and Rajan) across multiple generative models, with and without post-processing
Robustness tests against compression and downsizing
- Performance on challenging newer models like FLUX and aMUSEd
- The authors use appropriate controls, ensure diverse test datasets, and report standard metrics. The experiments effectively isolate the contribution of the Stay-Positive approach by comparing it with strong baselines using the same underlying architectures.

One particularly convincing aspect is the demonstration of improved performance on generative models not seen during training, which supports the claim about better generalization capabilities.

**Methods And Evaluation Criteria:**

The proposed methods and evaluation criteria are appropriate for the problem at hand. The Stay-Positive algorithm involves retraining the last layer of existing detectors to focus exclusively on fake image features while ignoring real image features, which directly addresses the identified problem of spurious correlations.

For evaluation, the authors use a diverse comprising:

- Real images from various sources (Redcaps, WikiArt, LAION-Aesthetics, whichfaceisreal)
- Fake images generated by multiple models (SDv1.5, MidJourney, Kandinsky, FLUX, etc.)
- Post-processed versions to simulate real-world conditions
- Partially Inpainted Images

The evaluation metrics (Average Precision) are standard and appropriate for binary classification tasks. The authors test across a wide range of scenarios, including different generative models and various post-processing operations, which provides a comprehensive assessment of the method's effectiveness and generalization capabilities.

**Other Comments Or Suggestions:**

Personally, I think the overall idea of "stay-positive" not only benefit the generated images detection. It would be helpful to discuss how the method might be extended to other media types (audio, video) or multimodal content.

- Consider including a more detailed analysis of cases where Stay-Positive underperforms or fails, which could provide insights for future improvements
- It would be beneficial to discuss the computational efficiency of the approach more explicitly, including additional training time and inference costs
- The paper could be strengthened by exploring more diverse post-processing operations beyond compression and resizing, for example, random noise, etc.

**Other Strengths And Weaknesses:**

Strengths:
- The proposed approach is conceptually simple yet effective, making it easy to implement on top of existing detectors
- The extensive evaluation across multiple generative models and post-processing techniques demonstrates the method's practical utility
- The focus on reducing spurious correlations represents a meaningful contribution to improving detector robustness

Weaknesses:
- The paper could benefit from a more detailed analysis of potential limitations or failure cases of the proposed approach
- There is limited discussion about the computational overhead or additional training time required for implementing Stay-Positive compared to traditional approaches

**Questions For Authors:**

See "Other comments" part

**Relation To Broader Scientific Literature:**

This work builds upon and extends previous research in fake image detection:

- It addresses limitations identified in recent works by Corvi et al. (2023) and Rajan et al. (2024), particularly regarding robustness to post-processing and generalization to new models.
- It connects to the broader literature on spurious correlations in machine learning models, applying these concepts specifically to the fake image detection domain.
- The paper relates to work on detecting images from diffusion models, flow-based models, and other generative architectures, extending detection capabilities to newer models like FLUX and aMUSEd.

**Theoretical Claims:**

The paper does not present formal mathematical proofs for theoretical claims. Instead, it focuses on empirical validation of the proposed approach through extensive experimentation. The conceptual foundation—that detectors should only focus on fake artifacts and ignore real image features—is well-articulated and supported by the experimental results, but no formal proofs are provided.

---

> ### Author Rebuttal · Authors · 2025-04-01
>
> **The paper could benefit from a more detailed analysis of potential limitations or failure cases of the proposed approach**\
> In this work, we have analyzed two limitations in detail, both related to the network's potential to learn spurious fake features. In Section 6, Fig 7, we show that our improved version of Corvi can still associate upsampling artifacts with the fake distribution. Additionally, in Appendix A.4, we explain a way in which the neural network (from stage-1 in Fig 3) could have learned to associate the absence of certain features with fake images, such as the lack of WEBP compression. We hope this clarifies the drawbacks of our work. We kindly request the reviewer to highlight any specific limitations that could be further explored, and we are happy to address them.
> \
> \
> **It would be beneficial to discuss the computational efficiency of the approach more explicitly, including additional training time and inference costs**
> \
> We thank the reviewer for raising these points. First, we would like to clarify that our method re-trains the final linear layer of the original network. Therefore, there are no additional inference costs, in comparison to the original method. We also compute the additional training time of our method (Rajan setting, only second stage), for which we use a batch size of 1024 on a single NVIDIA RTX A6000 machine. For optimal performance, we conduct stage-2 (Fig 3) re-training for 15 epochs which takes an additional 4h 8m 33s. Note that stage 1 takes 42h 5m 42s.
>
>
> **Personally, I think the overall idea of "stay-positive" not only benefit the generated images detection. It would be helpful to discuss how the method might be extended to other media types (audio, video) or multimodal content.**\
> We agree with the reviewer’s point. While the scope of our current work is focused on fake image detection, we believe the core principle of ignoring features associated with the real distribution will apply to other forms of media forensics, such as video and audio. We will address this in the final version of the paper.
>
> **The paper could be strengthened by exploring more diverse post-processing operations beyond compression and resizing, for example, random noise, etc.**\
> We thank the reviewer for raising this point, we have tested the sensitivity of our approach to JPEG Compression, additive gaussian noise and low-pass filtering based on the suggestions from various reviewers. We use the same experimental setting from Sec 3.1, where we take real images from reddit (Desai et al., 2021) and fake images from Stable Diffusion 1.5.
> \
> Link: https://imgur.com/a/HtYRCvO
> \
> We notice that both our Corvi+ and our Rajan+ are very robust to post-processing operations. This shows that our detector can reliably detect fake images in the wild.

---

> > ### Comment · Reviewer_JDtd · 2025-04-08
> >
> > Thanks for the rebuttal. I do not have any further question and will keep my score.

---

### Official Review · Reviewer_ndGS · 2025-03-15

**Overall Recommendation:** 2

**Summary:**

This paper proposed an algorithm designed to constrain the focus of detectors  to generative artifacts while disregarding those associated with real data. This method will help  the model reduce susceptibility to spurious correlations and enhance robustness.
## update after rebuttal
The authors have addressed some of my concerns, but I still have concerns about the quality of the paper, so I chose to give a weak rejection.

First of all, what confuses me the most is that the authors compared with an important existing method UFD, but as we all know, UFD also proposed a very influential public dataset UniversalFakeDetect, which includes 19 test settings, but the authors did not conduct experiments on UniversalFakeDetect but only on a subset CNN-generated images.

Secondly, Supplementary Material is Supplementary Material, and Appendix is Appendix. There is a special submission window for Supplementary Material, and authors can choose to provide their code, demonstration videos, or other materials. Of course, I read the Appendix provided by the authors, but this paper does NOT provide Supplementary Material.

In addition, the authors did not provide important ablation experiments at the beginning, and the newly added ablation experiment part cannot fully show the effectiveness and robustness of the proposed method.

Finally, although these will not affect my final score, I hope the authors can further improve the quality of writing. There are only ten quotation marks in the whole paper, and the authors only needs to search to find at least two errors (misuse of quotation marks: ”real” (line 402) and typographical errors: While most regions of such images are“real” (line 368) ). It is best for the authors to read the paper completely from beginning to end.

**Claims And Evidence:**

The authors' point that the distribution of real samples is harmful to detectors may need further discussion and verification. Many papers such as DIRE [1] and  [2] clearly state the benefits of positive samples and even think about the problem of generated image detection from a perspective similar to anomaly detection.

[1] Wang, Zhendong, et al. "Dire for diffusion-generated image detection." Proceedings of the IEEE/CVF International Conference on Computer Vision. 2023.

[2] Li, Jiaming, et al. "Frequency-aware discriminative feature learning supervised by single-center loss for face forgery detection." Proceedings of the IEEE/CVF conference on computer vision and pattern recognition. 2021.

**Essential References Not Discussed:**

No.

**Experimental Designs Or Analyses:**

The benchmarks provided by this paper are partially missing.

**Methods And Evaluation Criteria:**

Yes.

**Other Comments Or Suggestions:**

None.

**Other Strengths And Weaknesses:**

Advantages:
- The motivation of this paper is clear and the authors chose a straightforward but effective method to achieve the goal.
- The charts related to the experiments in the paper are relatively clear, and the organization of the charts is logical.

Disadvantages:
- Some of the images in the paper are not clear. Please provide images in PDF format if possible.

**Questions For Authors:**

- There are many larger benchmarks in the field of fake image detection, for example in DIRE [1] and UFD [3] there are larger benchmarks than the one in this paper, but the authors did not conduct a complete test.

[3] Ojha, Utkarsh, Yuheng Li, and Yong Jae Lee. "Towards universal fake image detectors that generalize across generative models." Proceedings of the IEEE/CVF Conference on Computer Vision and Pattern Recognition. 2023.

- The robustness tests in fake image detection also include the effects of JPEG compression and Gaussian noise. The authors should provide more robustness experimental tests.
- This paper does not provide ablation experiments to analyze the impact of different modules and parameters on the experimental results.
- The writing of this paper needs further improvement, for example, some quotation marks are incorrectly written
- Please unify the format of references. At least ensure that the citation formats of conferences and journals are consistent.
- In the paper, the authors show many examples that perform better than other methods. It would be better if the authors could show some visualization samples where the model fails to classify and analyze the reasons.

**Relation To Broader Scientific Literature:**

This paper may provide a new perspective in the field of generated image detection.

**Theoretical Claims:**

Yes.

---

> ### Author Rebuttal · Authors · 2025-04-01
>
> **The authors' point that the distribution of real samples is harmful to detectors may need further discussion and verification. Prior works show the benefits of “positive samples”**
> \
> We would like to make some clarifications. We are unsure of the exact meaning of 'positive samples,' but we assume it refers to real images. Our paper does not suggest that using real images for training harms the detector. In fact, we also incorporate real images in our training process. The key finding of our work is that associating specific patterns with the real distribution can often be spurious, which undermines the detector’s reliability.
> \
> \
> We provide concrete evidence for this claim. In Section 3.1, we show that the real distribution may contain unknown spurious correlations. Section 3.2 demonstrates that a detector trained to distinguish real images from LDM-generated ones can mistakenly associate certain features with real images. However, these same 'real' features appear in FLUX-generated images, proving the unreliability of 'real features' learned by discriminative models. Our results in Section 5 confirm this, as also noted by other reviewers (JDtd, GtjV, Qhvq).
> \
> \
> The DIRE detector suggested by the reviewer supports our argument by associating JPEG compression with real images, leading to spurious correlations, as discussed in AEROBLADE (Ricker et al., 2024) Section 9. Similarly, Work [2] attempts to reduce intra-class variability in real images but remains susceptible to the same issue. As shown in Section 3.2, the features defining 'realness' relative to a generator are often spurious.
> \
> \
> **The benchmarks provided by this paper are partially missing. There are larger benchmarks than the one in this paper.**\
> We seek clarification on what the reviewer means by “partially missing.” We assume this refers to the claim of not testing our method on popular public benchmarks. However, we respectfully disagree. As detailed in Appendix A.3, we evaluate our model on GenImage, a widely used, modern benchmark for fake image detection. Our GAN-based detector is also tested on the established CNNDet benchmark (line 329, Appendix A.5.2). While our primary benchmark is the publicly available dataset from Rajan et al. (2024), chosen for its inclusion of images from recent generators, we have also tested our LDM-based detector on the Autoregressive/Diffusion Models from the UFD benchmark, as suggested by the reviewer, the results can be found in https://imgur.com/a/eLTN1Kv. Our results show that the stay-positive algorithm improves detector performance across various unseen generators, hopefully addressing the reviewer’s concerns.\
> \
> **The robustness tests in fake image detection also include the effects of JPEG compression and Gaussian noise. The authors should provide more robustness experimental tests.**\
> Please refer to the response to reviewer JDtd.\
> \
> **Include Ablations**\
> We assume the reviewer refers to ablating other design choices for stay-positive. To do so, we conducted two ablations: (i) clamping detector weights without re-training, and (ii) re-training the entire network while clamping only the last layer.
>
> Results: https://imgur.com/a/H8fuyIZ
>
> Our results (similar to Table 2) show that clamping without re-training leads to suboptimal performance due to improper reweighting of fake features. Training the entire backbone while clamping the final layer underperforms on FLUX images, likely due to newly learnt spurious fake features. We hope this clarifies the reviewer’s concerns.
>
>
> **The writing of this paper needs further improvement, for example, some quotation marks are incorrectly written**\
> We're happy to refine the writing but request the reviewer specify which parts need improvement. The example regarding incorrect quotation marks is unclear, could the reviewer point to the relevant line or section?
>
> **Supplementary Material:
> None.**\
> We respectfully point out that this statement by the reviewer is incorrect. Our Appendix includes five sections that thoroughly discuss various details, and this has been acknowledged and confirmed by the other reviewers.
>
> **Please unify the format of references.**\
> We thank the reviewer for the suggestion and will unify the citation formats in the final version.\
> \
> **It would be better if the authors could show some visualization samples where the model fails to classify and analyze the reasons.**\
> In the Limitations section (Fig. 7), we show that our Corvi+ detector struggles with upsampled real images due to reliance on spurious features. Additionally, our models (Corvi+, Rajan+) fail to detect fake images from generators entirely different from the training distribution. Here are some qualitative examples:
>
> Firefly images (Detector trained on LDM images cannot detect these): https://imgur.com/a/GagThLg
> \
> Real Images and Upsampled Versions (Corvi, Corvi+ can detect the original 512x512 ones but cannot detect the upscaled 1024x1024 ones): https://imgur.com/a/MHX3Py0

---

### Decision · Program_Chairs · 2025-05-01

**Decision:**

Accept (poster)

**Comment:**

1x weak reject, 3x weak accept. This paper introduces an algorithm that constrains fake image detectors to focus exclusively on generative artifacts while ignoring real-image features, aiming to reduce spurious correlations and enhance robustness. The reviewers agree on the (1) clear motivation and simplicity of the proposed method, (2) comprehensive experimental evaluations and ablations that support improved robustness and generalization, and (3) effective empirical demonstrations across multiple generative models and post-processing scenarios. They also note (1) incomplete coverage of larger benchmarks and certain robustness tests (e.g., on UniversalFakeDetect, JPEG compression, and Gaussian noise), (2) some issues with writing clarity and image quality in the presentation, and (3) the potential amplification of spurious fake features due to retraining the last layer. The authors’ follow-up responses have convincingly addressed many of these concerns through additional experiments, clarifications, and expanded evaluations, so the AC leans to accept this submission.